# Resolution dependence of interlinked Southern Ocean biases in global coupled HadGEM3 models

David Storkey[1], Pierre Mathiot[2], Michael J. Bell[1], Dan Copsey[1], Catherine Guiavarc'h[1], Helene T. Hewitt[1], Jeff Ridley[1], and Malcolm J. Roberts[1]

[1]Met Office, FitzRoy Road, Exeter EX1 3PB, UK
[2]Univ. Grenoble Alpes/CNRS/IRD/G-INP/INRAE, Institut des Geosciences de l'Environnement, Grenoble, France

**Correspondence:** David Storkey (dave.storkey@metoffice.gov.uk)

**Abstract.** The early spin up of the HadGEM3 coupled model displays large-scale biases in the Southern Ocean at eddy-permitting ocean resolution: The subpolar gyres and Antarctic Slope Current (ASC) are too active; the Antarctic Circumpolar Current (ACC) transport is too weak; and there are large-scale water mass biases on the Antarctic shelf and in the open ocean. Most of the biases persist for at least 100 years of the model spin up. This set of biases is largely absent with a non-eddying ocean model and reduced with an eddy-rich ocean model. We show that damping the gyres and the ASC in the eddy-permitting model, either by introducing a parametrisation of baroclinic instability or by changing the lateral momentum boundary condition to increase bathymetric drag, acts to alleviate all the biases. This suggests that the fundamental issue in the eddy-permitting model may be to do with unresolved eddy processes and/or the representation of bathymetric drag on the flow. We investigate the structure of the biases in more detail and show that the eddy-permitting model has steep isopycnals near the Antarctic shelf slope consistent with a strong ASC and reduced transport of Circumpolar Deep Water (CDW) onto the shelf. However, across the region of the ACC jets the eddy-permitting model has shallower isopycnal slopes than the other models, consistent with a weaker ACC transport and warm near-surface biases in the open ocean.

## 1 Introduction

The Southern Ocean is a critical component of the Earth climate system which supplies a link between the main ocean basins via the Antarctic Circumpolar Current (ACC), between the near surface and deep ocean via the formation of Dense Shelf Water (DSW) and Antarctic Bottom Water (AABW), and between the near surface and mid-depth ocean via mode and intermediate water formation. It thus forms an important part of the global overturning circulation and helps to determine ocean uptake of anthropogenic heat and carbon (Marshall and Speer, 2012; Rintoul, 2018). Ocean processes in the Southern Ocean play a key role in the stability of ice sheets and hence are important for predictions of future sea level rise (Holland et al., 2010; Kusahara, 2020; Fox-Kemper et al., 2021).

The Southern Ocean is a particularly difficult region to model, due to its complex dynamics, including the interaction with the cryosphere. Historically, climate models have struggled to simulate large scale features of the the Southern Ocean accurately. For example, the upwelling of Circumpolar Deep Water (CDW) in low resolution models has been shown to be too slow (Drake et al., 2018) and large-scale warm biases at the surface are a common problem (Hyder et al., 2018). A key difficulty is that the Rossby radius of deformation, which sets the scale of geostrophic eddies, reduces to order 10 km or less at these latitudes (Chelton et al., 1998). Thus Southern Ocean eddies are unresolved or poorly resolved by the current generation of climate models, even those with higher ocean resolution (Hallberg, 2013; Hewitt et al., 2022). Eddies play a central role in setting the large-scale state of the Southern Ocean, for example in setting the momentum balance (Hughes and Ash, 2001), the vertical transfer of momentum (Marshall et al., 2017), the overturning circulation (Abernathey et al., 2011) and cross-shelf transports (Stewart and Thompson, 2015). Despite this, given the large computational cost of resolving eddies at high latitudes in global models, models in which eddies are unresolved or very poorly resolved are likely to continue to be used for multi-centennial and earth system modelling for the foreseeable future[1].

Hewitt et al. (2016) and Roberts et al. (2019) show that for the HadGEM3 family of coupled models (Williams et al., 2018), the simulation of the Southern Ocean appears to be particularly challenging at intermediate, or eddy-permitting ocean resolution. In a hierarchy of models with nominal 1° (non-eddying), 1/4° (eddy-permitting) and 1/12° (eddy-rich) ocean model resolution, the eddy-permitting model severely underestimates the ACC transport and has large warm sea surface temperature (SST) biases in the Southern Ocean. These biases are not present to the same degree in the non-eddying and eddy-rich models.

In this paper we investigate the biases in the hierarchy of HadGEM3 models in more detail. We show that there is a set of large-scale biases in the Southern Ocean which shows the same pattern across ocean resolutions, with large biases appearing at eddy-permitting resolution and then reducing again at eddy-rich resolution. As well as the weak ACC transport and warm SST biases discussed in previous papers, the subpolar gyres and Antarctic Slope Current (ASC) are too active in the eddying models, and cold, fresh biases develop on the Antarctic shelves. These biases all develop on similar timescales within the first 2-3 decades of the spin up, and, for the eddy-permitting model, tend to persist for at least 100 years of the model spin up.

The similarity in the cross-resolution pattern of the various biases in the initial spin up suggests that they are dynamically linked to each other. We investigate this by applying damping to the gyres and the ASC in the eddy-permitting model by using a scale-aware eddy parametrisation or changing the lateral momentum boundary condition to increase topographic drag, and look at the effect on the whole set of model biases. We examine the links between the biases in more detail, focussing on the link between the strong ASC and water mass biases on the Antarctic shelf, and the link between the weak ACC and the open ocean SST biases.

The paper is organised as follows. In Section 2, we describe in detail the Southern Ocean biases in the hierarchy of HadGEM3 models. In Section 3 we describe the results of the sensitivity experiments with the eddy-permitting model. In Section 4 we look at the density structure and the links between the different biases as well briefly discussing the energy balance of gyres and open ocean polynyas, and we summarise and conclude in Section 5.

---

[1]The development of ocean models with unstructured grids (Wang et al., 2014; Jungclaus et al., 2022) offers some potential for more efficient configurations which concentrate ocean resolution where it is needed.

|  | ORCA1 | ORCA025 | ORCA12 |
|---|---|---|---|
| **lateral viscosity** | laplacian $20000 m^2/s$ | bilaplacian $-1.5e+11 m^4/s$ | bilaplacian $-1.25e+10 m^4/s$ |
| **isopycnal tracer diffusion** | $1000 m^2/s$ | $150 m^2/s$ | $125 m^2/s$ |

**Table 1.**

Viscosity and isopycnal diffusion settings for the three ocean model resolutions. These values apply at the equator. The coefficients of laplacian viscosity and tracer diffusion reduce linearly with the grid spacing as the grid spacing reduces at higher latitudes. The coefficients of bilaplacian viscosity reduce with the cube of the grid spacing. This scaling is to avoid numerical instability.

## 2 Cross-resolution biases in the HighResMIP hierarchy

### 2.1 Description of experiments

We analyse integrations performed with the HadGEM3 GC3.1 configuration (Williams et al., 2018) following the HighResMIP protocol (Haarsma et al., 2016). The integrations are documented in Roberts et al. (2019). Full details of the HadGEM3 coupled model can be found in Williams et al. (2018) but for ease of reference we briefly summarise the ocean and sea ice models here. The ocean is based on the NEMO 3.6 code (Madec et al., 2019), set up with a z-star vertical coordinate system (Adcroft and Campin, 2004), 75 vertical levels with a resolution of 1m near the surface, and partial cells at the ocean bottom to better represent bathymetry (Barnier et al., 2006; Adcroft et al., 1997). Laplacian lateral viscosity is used in the non-eddying model and bilaplacian lateral viscosity is used in the eddying models with coefficients given in Table 1. A free slip lateral boundary condition on the momentum equation is used for all resolutions.[2] A parametrisation of baroclinic instability based on Tréguier et al. (1997) is used in the non-eddying model. In line with usual practice, no parametrisation of baroclinic instability is used in the two eddying models. Diffusion of tracers along isopycnal surfaces, parametrising eddy mixing, is used at all resolutions, with a coefficient that reduces at higher resolution (Table 1). The sea ice model is based on version 5.2.1 of the CICE model with multi-layer, energy-conserving thermodynamics (Bitz and Lipscomb, 1999), elastic-viscous-plastic ice rheology (Hunke and Dukowicz, 1997), and multi-category ice thickness (Bitz et al., 2001) with 5 categories. Cavities under ice shelves are closed and the output of basal melt water at the ice shelf front parametrised as described in Mathiot et al. (2017). A lagrangian iceberg model is used (Martin and Adcroft, 2010; Marsh et al., 2015). The distribution of ice shelf basal melt and iceberg calving are based on the seasonal observational estimates of Rignot et al. (2013) and Marsh et al. (2015) respectively. The total magnitude of basal melt and iceberg calving is set equal to the total precipitation falling on Antarctica at each timestep, ie. an assumption is made that the total mass of the Antarctic ice sheet is constant.

The HighResMIP experiment consists of a set of integrations with the ocean model at nominal 1°, 1/4° and 1/12° horizontal resolutions coupled in various combinations to the atmosphere model at N96, N216 and N512 resolution. After a short 30-year spin up ("spinup-1950"), two sets of experiments were performed: a constant 1950 forcing experiment ("control-1950") and an experiment with historical forcing ("hist-1950"). Here we are interested in the effect of ocean resolution so we analyse

---

[2]In the eddy-permitting model this changes to a partial slip condition around the coastline of Antarctica, and in the eddy-rich model it changes to a no-slip condition around the coastline of Antarctica. This was done to avoid instabilities associated with strong coastal currents.

| N216-ORCA1 | HighResMIP spin up (*spinup-1950* + *control-1950*) with ocean resolution of nominal 1° |
|---|---|
| N216-ORCA025 | HighResMIP spin up (*spinup-1950* + *control-1950*) with ocean resolution of nominal 1/4° |
| N216-ORCA12 | HighResMIP spin up (*spinup-1950* + *control-1950*) with ocean resolution of nominal 1/12° |
| N216-ORCA025-GM | HighResMIP spin up (*spinup-1950* + *control-1950*) with ocean resolution of nominal 1/4° with scale-aware Gent-McWilliams scheme |
| N216-ORCA025-PS | HighResMIP spin up (*spinup-1950* + *control-1950*) with ocean resolution of nominal 1/4° with partial slip south of 50°S |

**Table 2.**

Table summarising the integrations analysed. The ocean model uses the tripolar *ORCA* family of grids (Madec and Imbard, 1996), with *ORCA1* having 1° resolution at the equator and about 55km resolution at 60°S, *ORCA025* having 1/4° resolution at the equator and about 14km resolution at 60°S, and *ORCA12* having 1/12° resolution at the equator and about 5km resolution at 60°S.

the constant-forcing integrations with the N216 atmosphere coupled to the three ocean resolutions - ML, MM, and MH in the terminology of Roberts et al. (2019). For these resolutions the control-1950 integrations are a simple continuation of the spinup-1950 experiments so we treat these as a single spinup integration. The integrations analysed are summarised in Table 2. Initially we present results from the early spin up in the third decade and then show the longer term evolution.

Note that, as discussed in Section 4.3, neither the 1/4° model nor the 1/12° model fully resolve geostrophic eddies in the Southern Ocean. Nevertheless for clarity we have chosen to use the conventional description of these resolutions as "eddy-permitting" and "eddy-rich" respectively (as for example in Roberts et al. (2019)).

## 2.2 Biases in the third decade of spin up

Climate experimental protocols typically require models to be spun up for order centuries with pre-industrial forcing before the main experiments are performed to allow the model to come into quasi-equilibrium with the forcing and avoid the results of climate change experiments being contaminated with model adjustment processes. Due to the computational resources required to run high resolution coupled models, the HighResMIP protocol adopted a pragmatic approach in which the models were initialised with a 1950s climatology[3] and spun up for 30 years with 1950s forcing, then the main integrations were also short - of order 2-3 centuries. Roberts et al. (2019) argue that this is sufficient to show the impact of resolution on many aspects of the solution. We note that the bias metrics in Figure 7 appear to adjust rapidly over the first 2-3 decades of the integration and thereafter show much slower adjustment. We therefore first describe the main Southern Ocean biases based on time mean fields for years 21-30 of the spin up and then look at the time evolution.

The depth-integrated flow (Figure 1 top row) shows large differences in the strength and extent of the Southern Ocean subpolar gyres between the different ocean resolutions. For the non-eddying model the gyres are relatively weak and contained

---

[3]The climatology used to initialise the integrations and assess the model is a 1950-1954 mean of the EN4.2.2.g10 analysis (Good et al., 2013). Observations in the Southern Ocean are extremely sparse in this period, so in this region the 1950-1954 climatology is likely very similar to the background climatology used in the EN4 analysis which was for the period 1970-2000.

in their respective basins. The eddy-permitting model has much more active gyres (twice as strong as those in the non-eddying model) which have greater spatial extent and tend to merge into one another. The gyres in the eddy-rich model are still very active but slightly weaker and smaller in extent than in the eddy-permitting model. The westward flowing Antarctic Slope Current (ASC) is a nearly-circumpolar current and frontal zone that is associated with the Antarctic shelf break (Thompson et al., 2018). As well as having more active gyres, the higher resolution models also have a stronger ASC which in the eddy-permitting model is fully circumpolar, with a strong counterflow on the southern boundary of the Drake Passage, and westward flow along the shelf break in the Bellingshausen and Amundsen Seas associated with an eastward extension of the Ross gyre (Figure 2 top row). Observationally, the ASC in the Bellingshausen and Amundsen Seas is found to be very weak or even eastward-flowing (Thompson et al., 2018). 10-year mean depth-integrated transports for the Weddell and Ross gyres, as calculated by taking the spatial peak positive streamfunction value[4] are marked in Figure 1 for each of the three models. This is therefore a combined transport for the southern limb of the recirculating gyre and the ASC in each case. Klatt et al. (2005) found a transport of $56 \pm 8$ Sv for the southern limb of the Weddell gyre including the ASC[5], and Dotto et al. (2018) find $23 \pm 8$ Sv for the recirculating Ross gyre transport with an additional 6 Sv in the ASC. The gyres and ASC in the higher resolution models therefore seem to be considerably stronger than suggested by observations.

As noted by Hewitt et al. (2016) and Roberts et al. (2019), the ACC net transport through the Drake Passage is extremely weak in the eddy-permitting model with a time-mean model value of 90 Sv in the third decade of the integration compared to the estimate of 170 Sv due to Donohue et al. (2016). The Donohue et al. (2016) estimate is high compared to previous estimates (e.g. Cunningham et al. (2003) 136.7 Sv; Koenig et al. (2014) 141Sv) due to the inclusion of a large barotropic component, but the transport in the eddy-permitting model is very weak compared to these earlier estimates as well. The non-eddying model has a more reasonable value of 159 Sv and the eddy-rich model 119 Sv. The weaker ACC transport in the higher resolution models is associated with a flattening of the time-mean isopycnal slopes across the main part of the Drake Passage (Figure 3 top row). The eddy-permitting model has slightly steeper isopycnals at the northern edge of the strait associated with the main eastward jet and both of the higher resolution models have strong counterflowing currents, at the shelf break at the southern boundary, and associated with the Shackleton Fracture Zone in the centre of the strait. These counterflows significantly reduce the net transport. The counterflowing currents are stronger and more barotropic in the eddy-permitting model whereas in the eddy-rich model they are bottom-intensified. Xu et al. (2020) show bottom-intensified recirculations in the centre of the Drake Passage in a 1/12° model. They show a good match of the model to observations at mooring locations but argue that the observations undersample the recirculating currents. Meijers et al. (2016) observe a westward flowing extension of the ASC at the southern boundary of the Drake Passage with a magnitude of $1.5 \pm 1.5$ Sv - considerably smaller than that seen in the two higher resolution models.

---

[4]The streamfunction is calculated by integrating the velocity field northwards from the southern land boundary. The transports of the ASC and subpolar gyres therefore show as positive streamfunction values.

[5]Other estimates of the Weddell gyre strength are smaller, e.g. Yaremchuk et al. (1998) 34Sv and Reeve et al. (2019) 32Sv. In this paper we have chosen to compare with the Klatt et al. (2005) estimate as being most comparable to our Weddell gyre strength metric; it includes the transport over the shelf break which the Reeve et al. (2019) estimate does not, and the Klatt et al. (2005) section was located to the east of the gyre where the peak recirculating transport is.

The eddy-permitting model develops biases in the water mass properties on the Antarctic shelf. There is a tendency to freshening (Figure 4 top row) and cooling (Figure 5 top row) of the deep shelf waters compared to climatology in the two higher resolution models, this being most pronounced in the eddy-permitting model. There is a marked freshening of the deep waters for the whole Antarctic shelf in the eddy-permitting model, including for the regions of high-salinity shelf water (HSSW) in the western Weddell Sea and western Ross Sea (Mathiot et al., 2012). The eddy-rich model has freshening around the Antarctic Peninsula, the western Amundsen Sea and east Antarctica, but maintains (or intensifies) the regions of HSSW. Along the West Antarctic Peninsula and Bellingshausen and Amundsen Seas, circumpolar deep water (CDW) impinges on the shelf, giving relatively warm shelf water compared to the rest of the Antarctic shelf (Schmidtko et al., 2014). This warm water is still present in the non-eddying model after 30 years spin up, but is completely missing in the eddy-permitting model and partially eroded in the east of this region in the eddy-rich model (Figure 5).

There are resolution-dependent temperature biases in the open ocean which follow a similar pattern. The eddy-permitting model has significant warm SST biases in the region of the main ACC jets north of the subpolar gyres, particularly in the Indian Ocean sector east of the Kerguelan Plateau, and in the Pacific sector between the Ross Sea and Drake Passage (Figure 6 top row). The warm biases result in a reduced wintertime sea ice extent in this model. The warm biases are present, but reduced in magnitude in the eddy rich model. Both eddying models tend to have a slight cold bias in the subpolar gyres. The non-eddying model has a warm bias in the Pacific sector similar in magnitude to the eddy-rich model, but does not have a warm bias in the Indian Ocean sector, and the Southern Ocean SST overall tends to be too cold. The subsurface structure of these biases is discussed in Section 4.2.

## 2.3   Scalar metrics and time evolution of the biases

In order to examine the time evolution of the biases described in the previous section, we characterise them using scalar metrics (Figure 7). The maximum streamfunction values in the Weddell and Ross Seas are used to give the combined transport of the subpolar gyres and the ASC and compared with the observational estimates of Klatt et al. (2005) and Dotto et al. (2018) as described in the previous section. We calculate the net transport in the Drake Passage and compare to the Donohue et al. (2016) estimate. We also calculate the counterflow at the southern boundary of the Drake Passage as the total westward flow south of 62S and compare to Meijers et al. (2016) The temperature and salinity biases on the shelf are characterised by averaging the fields below 400m over relatively small areas where biases are indicative of poor representation of important processes: for the salinity biases, the areas of sea ice formation, brine rejection and deep water formation in the western Weddell Sea (WWED) and western Ross Sea (WROSS); and for the temperature biases, an area of the Amundsen sea (AMU) close to the front of the Pine Island Glacier and Thwaites Glacier ice shelves where the influx of Circumpolar Deep Water (CDW) onto the continental shelf is important for ice shelf dynamics. The WWED, WROSS and AMU areas are shown in the map in Figure 7. The volume-mean temperature and salinity values from the models are compared to time and spatial means of profile data from the EN4.2.2.g10 dataset (Good et al., 2013) as detailed in Appendix A. The open ocean SST biases are captured by averaging the model SST between 45S and 70S and comparing to a similar average performed on the 1950-1954 climatology of the EN4.1.1.g10 analysis dataset.

Comparison of the three HighResMIP integrations (solid lines in Figure 7) with the observational estimates (black dots and bars in Figure 7) shows that the transports of the gyres plus ASC in the eddying models spin up to large values, up to twice the observational estimates, in the first few decades. In the eddy-rich model the transports then decline over the next few decades to be within or close to the observational range, but the transports in the eddy-permitting model remain large on these timescales, especially in the Weddell Gyre. The non-eddying model has transports within or slightly below the observational range. The Drake Passage transport in the three models stabilises within the first few decades, with the non-eddying model slightly below the observed value of 170 Sv, the eddy-rich model too low at about 120 Sv and the eddy-permitting model very low at 80 Sv. The counter flow to the south of the Drake Passage in the eddying models also seems stable after the first 2 or 3 decades, albeit with large variability. For the deep salinities in the deep-water formation regions of the shelves, the salinities in the non-eddying and eddy-rich models stabilise at values at or above the observational estimates, indicating that the processes of sea ice formation and deep water formation may be being captured. But in the eddy-permitting model the salinities in these regions stabilise at a fresher value than observed (consistent with the the maps of 10-year mean fields shown in Figure 4), indicating that sea ice formation and deep water formation have to some extent been suppressed. For the region of CDW incursion in the Amundsen Sea, again the non-eddying and eddy-rich models appear to capture this, with temperatures being persistently warmer and within the observational range, whereas the eddy-permitting model quickly becomes cold and stays cold, indicating that the CDW is not intruding onto the shelf as it should or is being displaced by cold Weddell Sea water advecting around the Antarctic Peninsula (see discussion below).

For the preceding metrics the timeseries plots largely confirm that the 10-year mean maps presented in the previous section are representative of the behaviour of the model over the first century of spin up, with the exception of the subpolar gyres in eddy-rich model which start off too strong but then spin down. The maps of temperature and salinity biases on the shelves (Figures 4 and 5) give a wide area view of the biases whereas the corresponding scalar metrics focus on small areas. Thus the maps suggest that the non-eddying and eddy-rich models may be too fresh on the shelves in many places, but appear to capture the regions of deep water formation and high salinity in the western Weddell and Ross Seas as shown by the metrics, whereas in the eddy-permitting model the fresh biases include these regions, suggesting that deep water formation has been suppressed. The non-eddying model captures the relatively warm waters on the shelves in the Bellingshausen and Amundsen Seas, the eddy-permitting model is cold everywhere in this region, and the eddy-rich model has warm water in the west of the region but much colder in the easternmost part, suggesting that cold, fresh water may be advecting around the Antarctic Peninsula from the Weddell Sea. This is consistent with the strong westward flow at the southern edge of the Drake Passage in the eddying models. The Amundsen Sea metric is located close to the boundary between the cold and warm water in the eddy-rich model, so that it may show binary behaviour, flipping quickly between warm and cold states depending on how far west the cold water advances.

For the SST biases, the timeseries show that the biases in the eddy-permitting model evolve over the first century of the integration compared to what is shown in the 10-year mean maps. The warm biases in the Indian and Pacific sectors appear to spin down over the first century of the spin up so that the average SST bias over the whole Southern Ocean is close to the observational range but slightly too cold. The Southern Ocean average SST in the eddy-rich model starts within the

observational range and drifts slightly cold, suggesting that the warm biases in the ACC jets and the cold biases in the subpolar gyres (Figure 6) largely cancel in the spatial mean. The non-eddying model is significantly too cold in the spatial average.

## 3 Sensitivities in the eddy-permitting model

We have shown that the combined transport of the subpolar gyres and the ASC is too strong in the eddying models and this bias tends to persist in the eddy-permitting model. The large-scale ocean circulation is the result of a balance between the wind and buoyancy forcing from the atmosphere, and sinks of energy from the large-scale oceanic flow. The atmospheric resolution is the same for the three models so the explanation for the more active circulation must be due to the difference in ocean resolution and associated treatment of mesoscale eddies. In this section, we experiment with increasing the damping of the large-scale

circulation at high latitudes in the eddy-permitting model and look at the effect on the other biases.

We increase the damping in two ways. Firstly, we turn on the Gent-McWilliams scheme with a small coefficent in regions of small Rossby radius (ie. where eddies are poorly or not resolved). Tréguier et al. (1997) proposed a formula for the time- and space-dependent Gent-McWilliams coefficient based on dynamical constraints in a quasigeostrophic framework. We use a modified version of this scheme which is only applied in regions where eddies are deemed to be unresolved, as described in

detail in Appendix B. A typical spatial distribution of the resulting coefficient is shown in Figure 11. We note that the question of how to parametrise the effects of unresolved eddies in eddy-permitting models is an area of ongoing research (Hallberg, 2013; Jansen et al., 2019) and we are testing a fairly simple scheme here. Secondly, we change the lateral boundary condition on the momentum equations from a free-slip condition, where the shear next to bathymetry vanishes, to a partial-slip condition midway between the free-slip and no-slip cases (see Madec et al. (2019) section 8.1). This effectively increases the topographic

drag. For this experiment, partial slip was applied just in the Southern Ocean, south of 50S.

Results from the experiment with scale-aware Gent-McWilliams (N216-ORCA025-GM) and partial slip in the Southern Ocean (N216-ORCA025-PS) are shown alongside the results for the original HighResMIP hierarchy in Figures 1 - 7. Difference plots between the sensitivity runs and the control are shown for temperatures and salinities in Figures A1 - A3. For all the biases described here, the application of Gent-McWilliams or partial slip tends to ameliorate the bias to some extent. The

220 gyre strengths are reduced by 10-20% and their spatial extent reduced (Figures 1 and 7a-b). Looking at the timeseries (Figure 7), Gent-McWilliams appears to have a stronger impact than partial slip in the Ross Gyre, with the Gent-McWilliams test remaining about 10 Sv weaker than the control for the whole integration whereas the gyre in the partial slip test is about the same strength as the control after the gyre in the control weakens later in the run. The ACC transport is increased and the counterflows at the southern boundary and the Shackleton Fracture Zone are reduced (Figures 3 and 7c-d). In this case,

Gent-McWilliams and partial slip seem to have a similar impact on the net flow, increasing it by about 10 Sv. But the effect of Gent-McWilliams appears to be concentrated in reducing the counterflow at the southern boundary and changing the balance of eastward and westward jets, whereas partial slip reduces the jets across the Drake Passage (including the main eastward jet at the northern boundary to some extent).

The fresh biases on the shelves are reduced everywhere, with some recovery of the HSSW in the western Weddell Sea and western Ross Sea (Figures 4 and A1). The timeseries for the HSSW regions show that Gent-McWilliams appears to have a slightly stronger impact than partial slip in the western Ross Sea but Gent-McWilliams and partial slip have a similar impact in the western Weddell Sea. On the Amundsen shelf it is clear from the maps that partial slip has a stronger impact. The warm shelf in the Amundsen and Bellingshausen Seas is recovered to some extent in the sensitivity runs (Figures 5 and A2). As for the salinity in this region, the partial-slip test shows a stronger sensitivity than the Gent-McWilliams test. The timeseries show evidence of the binary behaviour associated with the advance or retreat of cold water from the east discussed in Section 2.3: the control moves quickly to a cold state, the partial-slip test stays warm, and the Gent-McWilliams test becomes cold but then flips back to a warm state. The stronger sensitivity of this metric to the partial slip test is consistent with partial slip having a greater damping effect on the westward counterflow at the southern boundary of the Drake Passage (Figure 3).

The warm SST biases in the first part of the spin up of the eddy-permitting model are reduced the Indian Ocean sector in both cases (Figures 6 and A3). However, the other main region of warm bias in the Pacific sector west of the Drake Passage is warmed slightly in the partial slip test and cooled by Gent-McWilliams. The wintertime sea ice extent responds as expected to changes in SST bias, expanding where the SST has cooled and retreating where it warms. The SST bias metric (Figure 7h) is consistent with the maps, showing that Gent-McWilliams has a greater overall cooling effect in the Southern Ocean than partial slip. Overall the spatial mean SST in the Gent-McWilliams test tends to become too cold over the course of the 100-year integration, whereas the partial slip test parallels the control for the second half of the integration.

## 4 Discussion

### 4.1 Cross-shelf density structure

We have shown that the resolution dependence of the bias in the ASC transport and the bias in the water mass properties on the Antarctic shelf tend to follow the same pattern, with the largest biases in the eddy-permitting ocean model. In this section we look at the link between these two sets of biases in more detail. The properties of the shelf water are controlled partly by local surface fluxes and associated water mass transformations and partly by the exchange of water with the open ocean across the shelf break. In particular, the extent to which Circumpolar Deep Water (CDW) is transported onto the shelf is crucial (Schmidtko et al., 2014), and this is mediated by the structure of the Antarctic Slope Front associated with the ASC.

The review of the ASC by Thompson et al. (2018) identifies three distinct regimes: in their terminology, the Fresh Shelf, the Dense Shelf and the Warm Shelf. In Figure 8 we compare sections from the models to the hydrographic data plotted by Thompson et al. (2018) to shed some light on the links between the biases described in the previous sections. The approximate locations of the three sections are shown in Figures 4 and 5.

For the Fresh Shelf section in the eastern Weddell Sea (Figure 8, left-hand column), the isopycnals tend to align with isotherms and slope steeply downwards towards the pole, incropping to the shelf slope, and producing a strong horizontal density gradient associated with a vigorous ASC. The strong front acts as a barrier to incursion of CDW onto the shelf; these incursions are likely to only happen occasionally as tidal or eddy driven fluctuations of the front position onto the shelf (Wang

et al., 2013; Goddard et al., 2017). All the models capture the large scale pattern in this regime with downward sloping, incropping isopycnals. The isopycnal slopes in the two higher-resolution models appear to match the observations quite well, but the stratification is greater than observed in the eddy-permitting model. This appears to be due to a greater entrainment of CDW water into the gyre giving warmer and saltier water at depth. The shelf is narrow in this region and is barely resolved in the non-eddying model.

In the Dense Shelf section in the western Weddell Sea (Figure 8, middle column) the observed structure is more complex, with a V-shaped pattern of isopycnals associated with the incursion of CDW onto the shelf and its transformation and export as Dense Shelf Water (DSW). The V-shape is not present in the non-eddying model, which does not have the resolution to capture this structure. It is present to some extent in the both the higher-resolution models but the overall structure is much better captured in the eddy-rich model. The deeper isopycnals in the eddy-permitting model slope down towards the shelf slope and incrop, and the dense overflow water is completely lost. Similarly to the case for the Fresh Shelf section, the stratification is stronger in the eddy-permitting model with warmer subsurface waters.

On the Warm Shelf in the Bellingshausen Sea (Figure 8 right-hand column) the isopycnals slope upwards towards the shelf and allow along-isopycnal mixing of CDW onto the shelf. This structure is well captured by the non-eddying and eddy-rich models but the eddy-permitting model again has downward sloping and incropping isopycnals in this region associated with a strong westward flowing ASC (see Figure 2) which will act as a barrier to the mixing of CDW onto the shelf, and the shelf is colder than in the other models with no signature of CDW.

In general, one can say that the shelf slope region is poorly resolved by the non-eddying model. The eddy-permitting model tends to be more stratified and have isopycnals that slope more steeply towards the shelf slope, consistent with a stronger westward flowing ASC. This pattern is the same as seen on the southern boundary of the Drake Passage in Figure 3. The stronger frontal zone will act as a barrier to the mixing of warm, salty CDW onto the shelf and help to explain the link between over-active gyres and ASC and the cold, fresh biases on the shelf. The strong ASC will also have a tendency to advect cold, fresh Weddell water around the Antarctic Peninsula into the Bellingshausen and Amnundsen Seas.

## 4.2 Open ocean temperature biases and slumping of isopycnals

As discussed in Section 2.2, warm SST biases develop in the eddying models in regions of the main ACC jets. In Figure 9 we show the subsurface structure in one of the regions of the largest bias, along 90°E, downstream of the Kerguelan Plateau. (The location of the section is marked in Figure 6). The two high resolution models show warm biases in this region, particularly in the top 400m between 65S and 50S and extending northward subsurface along isopycnals. The biases show a similar pattern in the two models but are more intense in the eddy-permitting model. By contrast the non-eddying model shows a slight cold bias near the surface. The large-scale structure of the isopycnals is different between the models, with the high resolution models showing a slumping of the isopycnal slopes across the section which is most pronounced in the eddy-permitting model. For example the 27.15 isopycnal outcrops at about 60°S in the eddy-permitting model and at about 57°S in the eddy-rich model. The isopycnals are steeper, particularly around 50°S, in the non-eddying model and the EN4.1 climatology. Conversely, the higher resolution models show the steepest isopycnals near the continent, associated with a more active ASC. The EN4.1 climatology

does not show downward sloping isopycnals near the continent in this region. This is likely due to the low resolution of the analysis and the sparseness of the included observations. Peña-Molino et al. (2016) show steeply sloping isopycnals associated with a westward flowing ASC at 113°E. The overall pattern, with the higher resolution models showing slumping isopycnals across the ACC jet and steeper isopycnals near the continent associated with a strong westward flow, is similar to the pattern

of biases seen in the Drake Passage (Figure 3). The use of scale-aware Gent-McWilliams or partial slip in the eddy-permitting model damps the ASC and reduces the isopycnal slopes near the continental shelf break, and also reduces the slumping of isopycnals and the associated warm biases in the open ocean (Figure A4).

The section plots suggest a possible relationship between the density structure and the temperature biases in this region. Because the density variations are controlled more by salinity at high latitues, eddies tend to mix heat upwards along isopycnals

across the ACC (Gregory, 2000). In the models this process is achieved both by diffusion along isopycnal surfaces and by resolved eddies. (As noted above, the higher resolution models include isopycnal diffusion with a reduced coefficient compared to that used in the non-eddying model). Given the same amount of isopycnal mixing, slumping of the isopycnals will tend to allow heat to be transported further south which could explain increased near-surface warm biases in the models with shallower isopycnal slopes.

The fact that the use of Gent-McWilliams or partial slip tends to steepen the isopycnal slopes in the open ocean is somewhat unexpected. A possible explanation involves the enhanced mixing due to truncation errors in advection schemes which is known to exist in z-coordinate models. Lee et al. (2002) show that an eddy-permitting model tends to rapidly lose the densest waters in the Southern Ocean due to spurious diapycnal mixing from the tracer advection scheme, particularly under high frequency forcing. The loss of dense water results in a sinking of isopycnal surfaces, which resembles the large-scale slumping of the

isopycnals seen in the current models. Ilıcak et al. (2012) show that spurious diapycnal mixing can be controlled by limiting the gridscale Reynolds number. It may be that damping the subpolar gyres and the ASC reduces spurious mixing in the model by this mechanism, thus tending to maintain stronger fronts across the ACC. The investigation of this possibility will be the subject of future work.

### 4.3 Damping of gyres and ASC

In the first few decades of the spin up of the eddy-permitting and eddy-rich models the subpolar gyres increase in strength quickly before plateauing and, in some cases, slowly declining (Figure 7). In equilibrium there must be a balance between the energy input to the large scale ocean flow by wind stress and buoyancy fluxes and the net energy flux to smaller scales. For the present case we are using the same atmosphere model across the different ocean resolutions and we might expect the large-scale wind forcing to be similar. Figure 10 shows the time-mean zonal winds at 10m for the three models compared with

a mean from the JRA-55 reanalysis (Kobayashi et al., 2015). The large-scale patterns are very similar, as is the large-scale pattern of wind stress curl (not shown). The model winds are generally a bit weaker than the reanalysis, probably because N216 is only medium resolution in the atmosphere[6]. The fact that the model winds at large scales are a bit weaker than the reanalysis winds tends to reinforce the argument that the unrealistic spin up of the gyres in the eddying models is not due to

---

[6]N216 is roughly equivalent to 60km resolution in the atmosphere.

the wind forcing. The buoyancy forcing will also be similar across the models in the initial stages of the spin up, but as noted by Beadling et al. (2022), a sufficiently strong ASC can block the export of fresh water from the shelf region, resulting in a positive feedback whereby the build up of fresh water on the shelf creates a stronger off-shelf density gradient, resulting in a stronger ASC.

A major sink of energy from the large scale flow is thought to be the generation of mesoscale eddies through baroclinic instability (Wunsch and Ferrari, 2004). Jamet et al. (2021) use multi-scale analysis and model results to show that the production of eddies in the coastal and separated boundary currents is an important sink of energy from the large-scale flow in the North Atlantic subtropical gyre. The Rossby radius of deformation is not resolved by the eddy-permitting model at the latitudes of the southern subpolar gyres and only barely resolved by the eddy-rich model (Hallberg, 2013). So it is clear that eddy processes will be poorly or not represented in these regions and this sink of energy from the large scale flow will be missing. The result with the scale-aware Gent-McWilliams scheme indicate that adding a slumping of isopycnals within the subpolar gyres tends to improve the solution in multiple ways.

The HadGEM3 model uses a z-level vertical coordinate with partial cells permitted next to bathymetry (Barnier et al., 2006). As an artifact of the formulation, topographic drag in z-level models is split between drag on the bottom of the cells and drag on the side walls, the former being formulated using as a nonlinear drag, and the latter as a lateral boundary condition on the momentum equation. Here we have chosen to alter the lateral boundary condition rather than the bottom boundary condition. This will tend to have a greater impact where the bathymetry is steep. Topographic drag in the model represents the effect of unresolved processes in the bottom boundary layer such as lee wave drag (Garabato et al., 2013). While topographic drag appears to be a small term in the vorticity balance (Garabato et al., 2013; Styles et al., 2022), changes in this term will still affect the overall balance and a representative value for the deep ocean is uncertain. Furthermore, the z-level formulation is not a natural way to represent the bathymetry and it is possible that some of the flow-bathymetry interactions arising in this form of the model are spurious (Styles et al., 2022). Here we have shown that there is a significant sensitivity to varying the topographic drag at least at eddy-permitting resolution, and for the Southern Ocean the effects can be far-ranging.

## 4.4 Open ocean polynyas

Many climate models, including HadGEM3, exhibit open ocean polynyas and associated deep convection in the Weddell and Ross Seas. Menary et al. (2018) describe such polynyas opening up in the eddy-permitting version of the HadGEM3 model which then drastically affect the density structure of the central Weddell Sea. Behrens et al. (2016) show that much of the decadal variability in the Southern Ocean in CMIP models appears to be driven by the variability of these deep convection events. For the integrations examined here, open ocean deep convection in the Weddell Sea first occurs 50 years into the spin up in the eddy-permitting and eddy-rich models and intermittently thereafter (not shown). Therefore the polynyas do not form part of the causal chain of events leading to the biases examined in this paper, which are fully developed after three decades. However, Behrens et al. (2016) show that stronger gyres and associated increased offshore freshwater export by ocean currents can act as a precursor to deep convective events, so it is possible that the same mechanism is happening here, with the biases in the early spin up, particularly the strong gyres, acting as precursors to the occurence of the polynyas.

## 5 Conclusions

We have investigated the pattern of biases in the Southern Ocean in the HadGEM3 family of coupled models and shown that the model with eddy-permitting ocean resolution displays a series of biases which all spin up on timescales of 2 to 3 decades and largely persist for the first 100 years of the spin up: the subpolar gyres and ASC are too active, the ACC transport in the Drake Passage is too weak, the water masses on the Antarctic shelves tend to be too cold and fresh, and there are near-surface warm biases in regions of the main ACC jets. These biases are largely absent from the non-eddying model and reduced in the eddy-rich model. Applying damping focussed at high latitudes, either by applying a scale-aware version of the Gent-McWilliams eddy parametrisation with a small coefficient, or by changing the lateral boundary condition on momentum from free-slip to partial slip reduces all the biases to some extent. In general the impact of the two model changes seems to have a similar impact, although partial slip has a slightly larger impact on the Weddell gyre strength, counterflow in the Drake Passage and the Amundsen/Bellingshausen temperature and salinity biases. We have suggested that the poor representation of eddy processes at these latitudes and/or poor representation of the topographic influence on the flow may be a part of the explanation for the set of biases in the eddy-permitting model.

We have investigated the structure of the biases in more detail and shown that near the shelf break the eddy-permitting model tends to be more stratified than the other models and to have isopycnals that slope more steeply towards the shelf slope, consistent with a stronger ASC. The stronger front in the eddy-permitting model acts as a barrier to the inflow of warm, salty Circumpolar Deep Water onto the shelf and to the export of cold, fresh water off the shelf; thus the shelf waters in the eddy-permitting model tend to be too cold and fresh.

In the open ocean, isopycnals tend to have shallower slopes across the ACC jets in the eddy-permitting and eddy-rich models compared to the non-eddying model and the climatology. The weaker horizontal density fronts are consistent with the weaker ACC transport seen in these models. The shallower open-ocean isopycnal slopes may also explain the near-surface warm biases seen in some regions of the ACC, with isopycnals outcropping further south in the high resolution models than in the non-eddying model and permitting heat to be transported further south by eddy processes (resolved or parametrised).

In this paper, we have focussed on a subset of the integrations in Roberts et al. (2019) with the three ocean resolutions coupled to an N216 atmosphere, in order to focus cleanly on the differences in behaviour due to differences in ocean resolution and parametrisations. Roberts et al. (2019) show that some of the patterns in biases across ocean resolution seem relatively insensitive to atmosphere resolution. In particular, their Figure 18 shows that the very weak ACC at eddy-permitting ocean resolution is unaffected by atmosphere resolution. We have also focussed on trying to reduce the biases in the eddy-permitting model since this has the largest Southern Ocean biases. Tests with scale-aware Gent-McWilliams and partial slip in a forced integration of the ORCA12 model (not shown) have shown a small reduction in the Southern Ocean biases, in particular a slight increase in the net Drake Passage transport,

The results with coupled GCMs presented in this paper show similarities to results presented by Styles et al. (2023) in an idealised ocean-only model of the Weddell gyre run at a range of horizontal resolutions. When they use a bathymetry with some artificial roughness added (as opposed to a simple box model), they find that the model with eddy-permitting resolution

shows the steepest isopycnals and the most active gyre. They attribute the shallower isopycnals in the other resolutions to the use of Gent-McWilliams at non-eddying resolution and the action of explicitly resolved eddies in the eddy-rich resolution. They find that using Gent-McWilliams at eddy-permitting resolution shallows the isopycnal slopes as expected, and reduces the gyre strength.

There is a general pattern in the structure of the isopycnals with the eddy-permitting model showing steeper isopycnal slopes near the shelf slope around Antarctica and shallower isopycnal slopes across the ACC jets. Application of damping focussed at the highest latitudes tends to reverse both of these trends (see for example Figure 3). It is straightforward to see how the application of Gent-McWilliams and increased topographic drag could damp boundary currents and reduce the slope of isopycnals near topography. But as we noted in Section 4.2, it is less obvious why these changes result in reduced slumping (ie. steepening) of isopycnal slopes in the open ocean. We have suggested a possible mechanism involving the loss of AABW due to spurious numerical mixing. It is also clear (Figure 8) that the representation of the formation process of AABW due to the creation and export of dense water from the shelves is not well represented by the non-eddying and eddy-permitting models. A detailed study of the different models' representation of the dense water reservoir and the relationship to the biases described in this paper will be the subject of future work.

*Code availability.* TEXT

*Data availability.* TEXT

*Code and data availability.* The ocean and sea ice model code is available at Storkey (2024a). Data and scripts to produce the plots in the paper are available at Storkey (2024b) and Storkey (2024c).

**Appendix A: Processing of EN4 profile data for comparison with model data on the Antarctic shelves**

As described in Section 2.3, volume-mean deep temperature and salinity values are used to characterise the evolution of model biases on the Antarctic shelves. For the observational comparator, profiles from the EN4.2.2.g10 dataset were used (Good et al., 2013). For each region, all the available profiles with an overall quality control value of 1, for the full timeseries between 1900 and 2021, were used to create means over the relevant areas and depth ranges. For the western Weddell Sea the available profiles are mainly cruise data, heavily biased towards the austral summer months. For this region we use summertime (DJF) means of the model data to better match the available observations. For the western Ross Sea and Amundsen Sea areas, there are marine mammal observations (McMahon et al., 2021) available for recent years with good year-round coverage, so for these areas we use annual mean model data. These observations also have good depth coverage since Weddell seals regularly

dive to up to 700m depth. The data coverage, while good given the remoteness of the region, is still spatially and temporally sparse. To avoid biases due to particular seasons and depths being better sampled, the time and depth averaging was done in monthly and 100m bins first and these then averaged to produce the final mean values.

## Appendix B: Formulation of the scale aware Gent-McWilliams scheme

The Gent-McWilliams scheme in the non-eddying model uses a 2D spatially- and temporally-varying coefficient $\kappa$, due to Tréguier et al. (1997), based on the scaling arguments of Held and Larichev (1996), where $\kappa$ is defined as

$$\kappa = \frac{Ro^2}{T_{eff}}. \tag{B1}$$

Here $Ro$ is the first internal Rossby radius of deformation,

$$Ro = \int N dz \Big/ \pi . f_0 , \tag{B2}$$

where $N$ is the Brunt-Väisälä frequency and $f_0$ is Coriolis parameter. $T_{eff}$ is a timescale for the growth of baroclinic instabilities,

$$T_{eff}^2 = \int N^2 . (S_x^2 + S_y^2) dz \tag{B3}$$

with $S_x$ and $S_y$ the slopes of isopycnals in the $x$- and $y$- directions.

For the experiment with the eddy-permitting model we impose a cap $\kappa_{max}$ such that $\kappa < \kappa_{max}$ with $\kappa_{max}$ varying with $Ro$ as follows:

$$\kappa_{max} = min(1.0, \frac{2}{3} * (2.0 - Ro/\Delta x)) * 75.0 m^2/s, \tag{B4}$$

where $\Delta x$ is the horizontal grid spacing. So $\kappa_{max}$ ramps up linearly from zero in regions where eddies are deemed to be resolved ($Ro/\Delta x > 2$) to a weak value of $75.0 m^2/s$ where eddies are not resolved ($Ro/\Delta x = \frac{1}{2}$). Figure (11) shows an example of the typical Gent-McWilliams diffusion coefficients produced in practice.

A value of $75.0 m^2/s$ for the diffusion coefficient is very small compared to typical values of order $1000.0 m^2/s$ used in non-eddying models. The reason for choosing such a small value was largely pragmatic; much larger values were found to degrade the model solution in other regions, particularly the North Atlantic. The question of how best to parametrise the effects of eddies in models where the eddies are partially resolved is a topic of ongoing research (eg. Hallberg (2013); Jansen et al. (2019)).

*Author contributions.* MJR designed and carried out the HighResMIP experiments. DS carried out the sensitivity experiments with the eddy-permitting model. PM designed the scalar metrics described in section 2.3 and wrote the associated code. All authors were part of the Southern Ocean Process Evaluation Group to try to understand the biases in the eddy permitting model. DS wrote the manuscript with input from all co-authors.

*Competing interests.* The contact author has declared that none of the authors has any competing interests.

*Disclaimer.* TEXT

*Acknowledgements.* The authors would like to acknowledge discussions with Pat Hyder who led the Southern Ocean Process Evaluation
Group. We would also like to thank Malin Ödalen and an anonymous reviewer whose detailed comments have helped to improve the
manuscript.

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

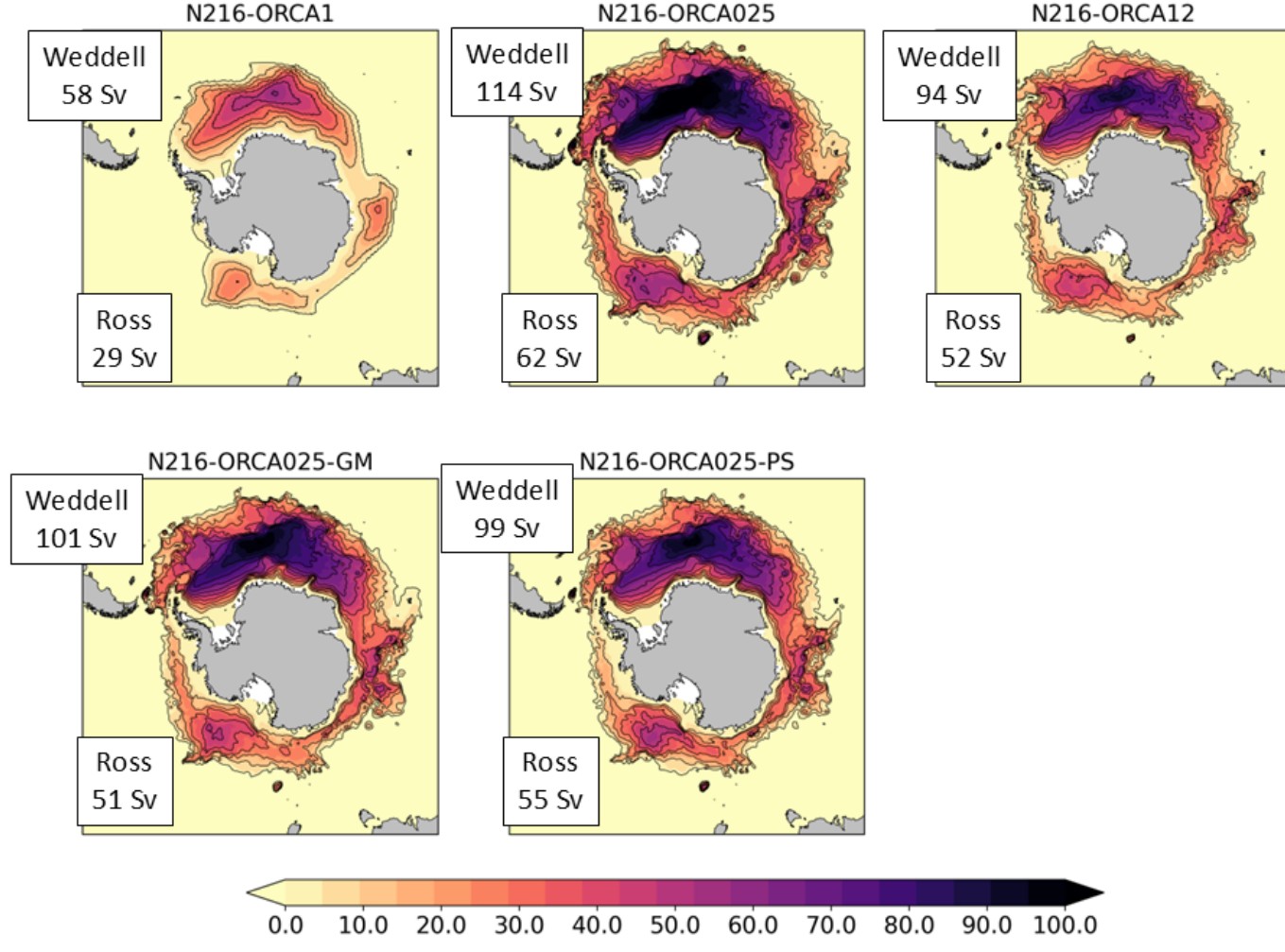

**Figure 1.** The barotropic streamfunction (Sv), time mean for the third decade of the spin up. The experiment labels are defined in Table 2. The integration of the velocity field is done from the south northwards and only positive streamfunction values are plotted, highlighting the subpolar gyres south of the main ACC fronts. The spatial peak of the time-mean streamfunction values for the gyres in the Weddell Sea and Ross Sea are marked. By construction this includes the transport of the southern limb of the recirculating gyre plus the transport of the ASC. These numbers can be compared to the observational estimates of $56 \pm 8$ Sv (Klatt et al., 2005) for the Weddell gyre, and $29 \pm 8$ Sv (Dotto et al., 2018) for the Ross gyre.

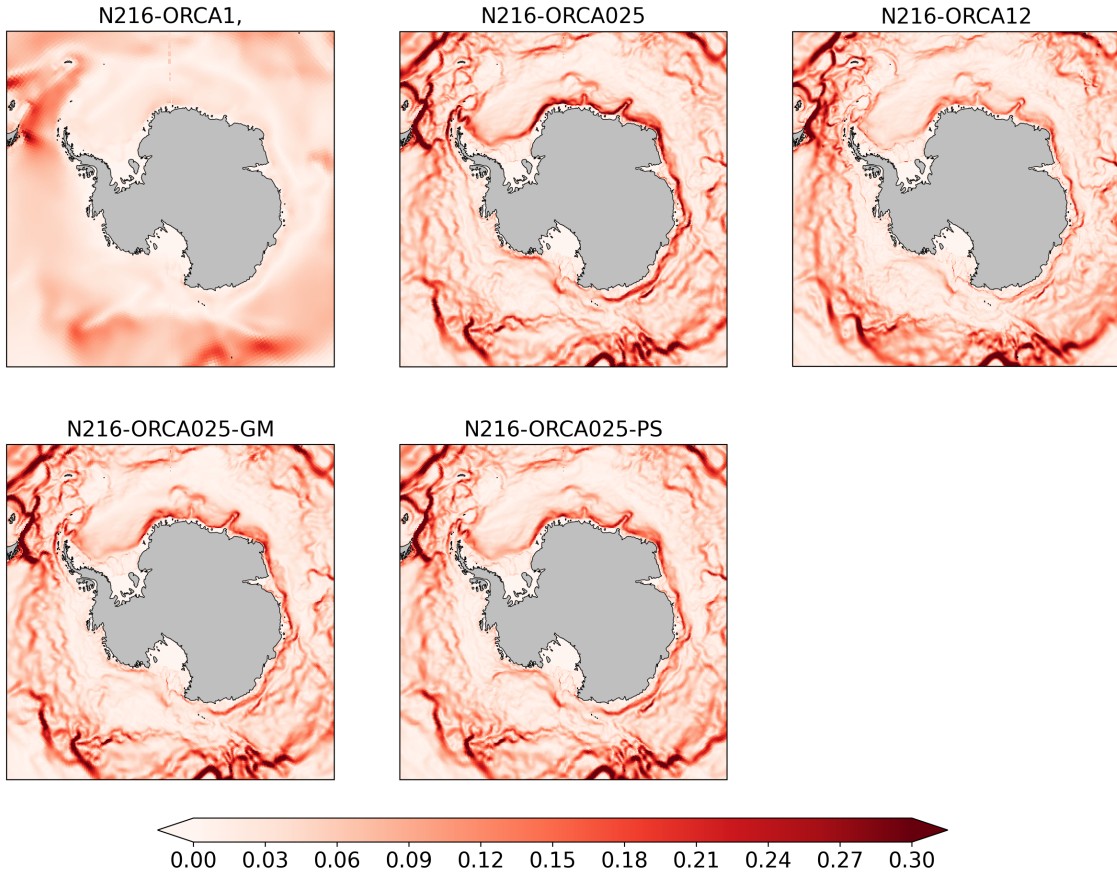

**Figure 2.** Depth-mean current speeds (m/s) over the top 500m, time mean for the third decade of the spin up. The experiment labels are defined in Table 2. The eddying models show a strong westward flowing slope current, which in the eddy-permitting model is fully circumpolar, including flow at the southern boundary of the Drake Passage and in the Bellingshausen and Amundsen Seas. Note in the region of the gyres the flow along the slope is a combination of the recirculating flow in the gyres and the throughflow.

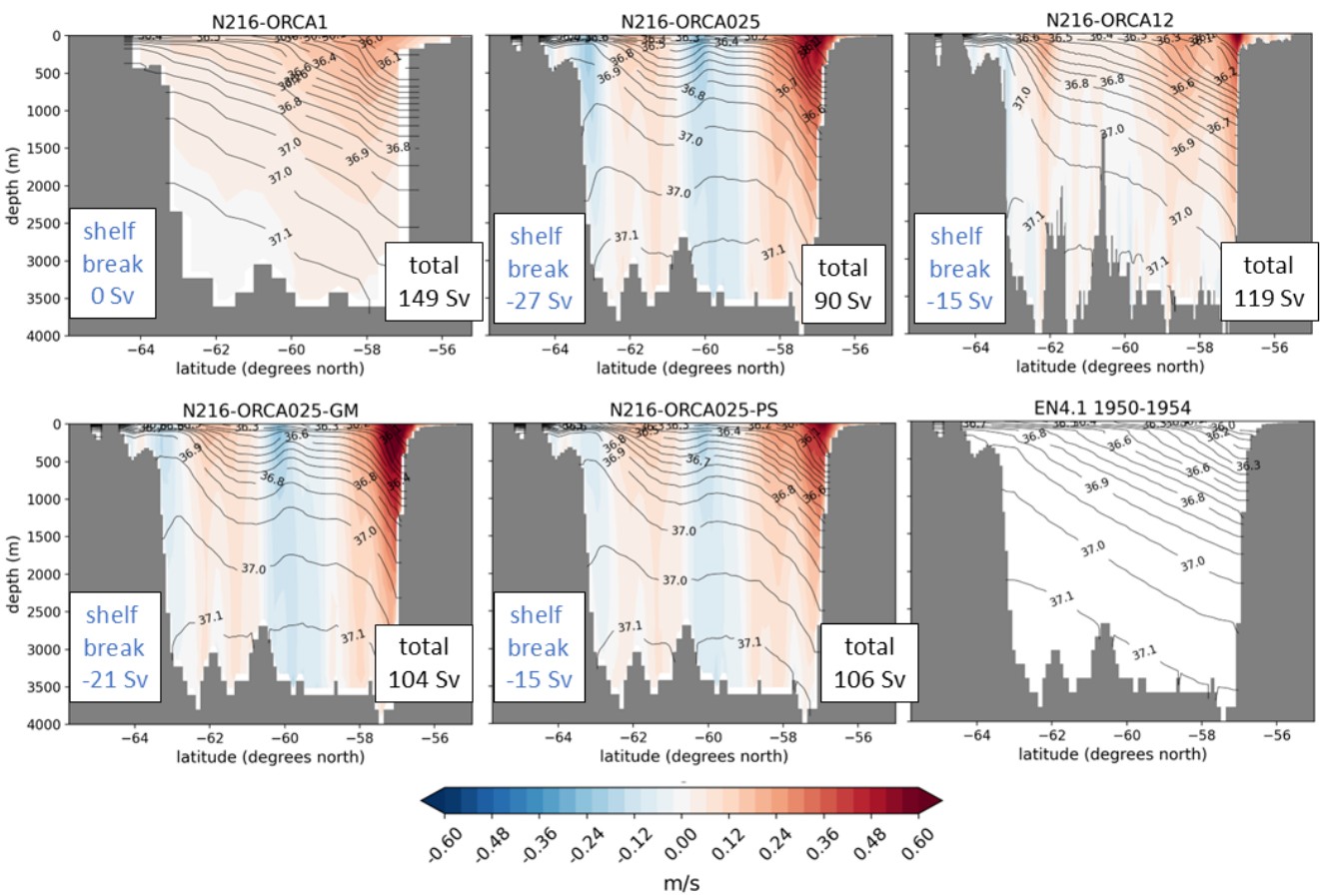

**Figure 3.** Cross section at the Drake Passage of zonal velocity (colours) and potential density with respect to 2000m ($\sigma_2$ - lines). Time mean for the third decade of the spin up. The experiment labels are defined in Table 2. The net transport ("total") and the counterflow transport next to the shelf break at the southern boundary ("shelf break") are marked. The net transport can be compared to the Donohue et al. (2016) estimate of 170 Sv. The shelf break transport is defined as all westward flow south of 62°S. Meijers et al. (2016) observe a westward flow at the southern boundary of the Drake Passage with a magnitude of $1.5 \pm 1.5$ Sv. Isopycnals from a 1950-1954 climatology of the EN4.1 reanalysis (Good et al., 2013) are shown in the bottom right figure.

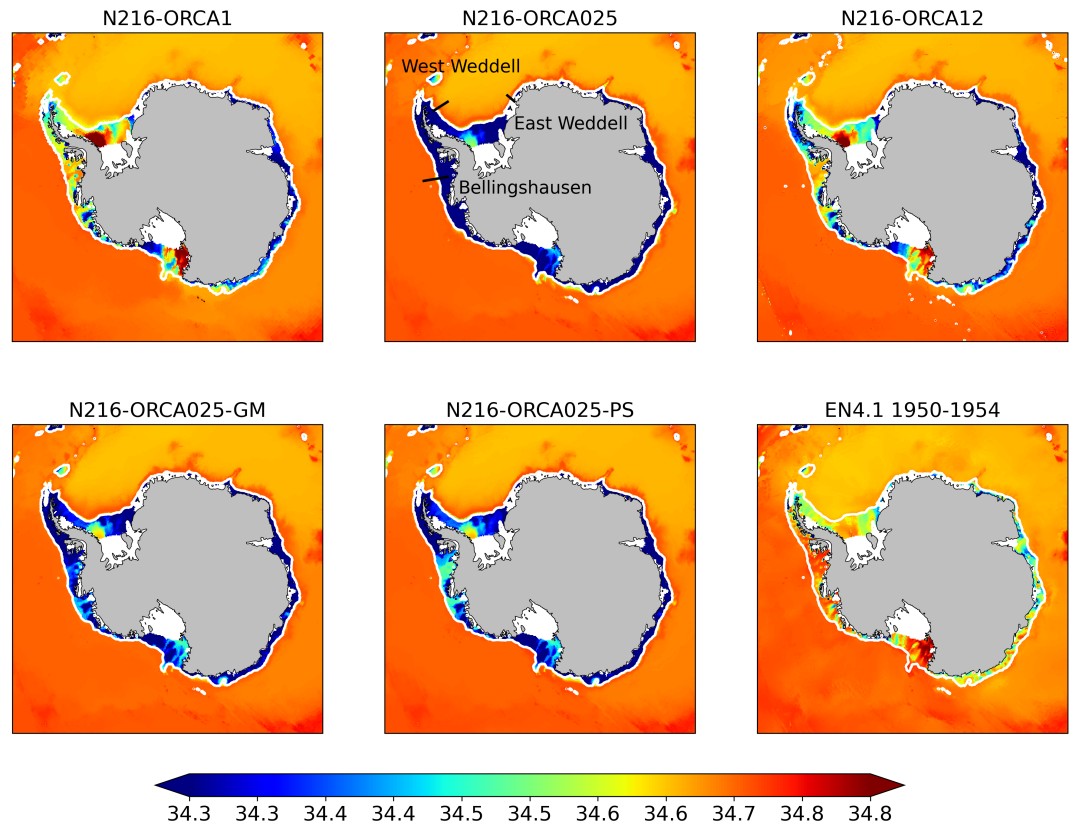

**Figure 4.** Bottom salinities (psu), time mean for the third decade of the spin up. The experiment labels are defined in Table 2. The 1000m depth contour is shown in white. Bottom temperatures from a 1950-1954 climatology of the EN4.1 reanalysis (Good et al., 2013) are shown in the bottom right figure. The top middle figure shows the approximate locations of the sections described in Thompson et al. (2018) and plotted in Figure 8.

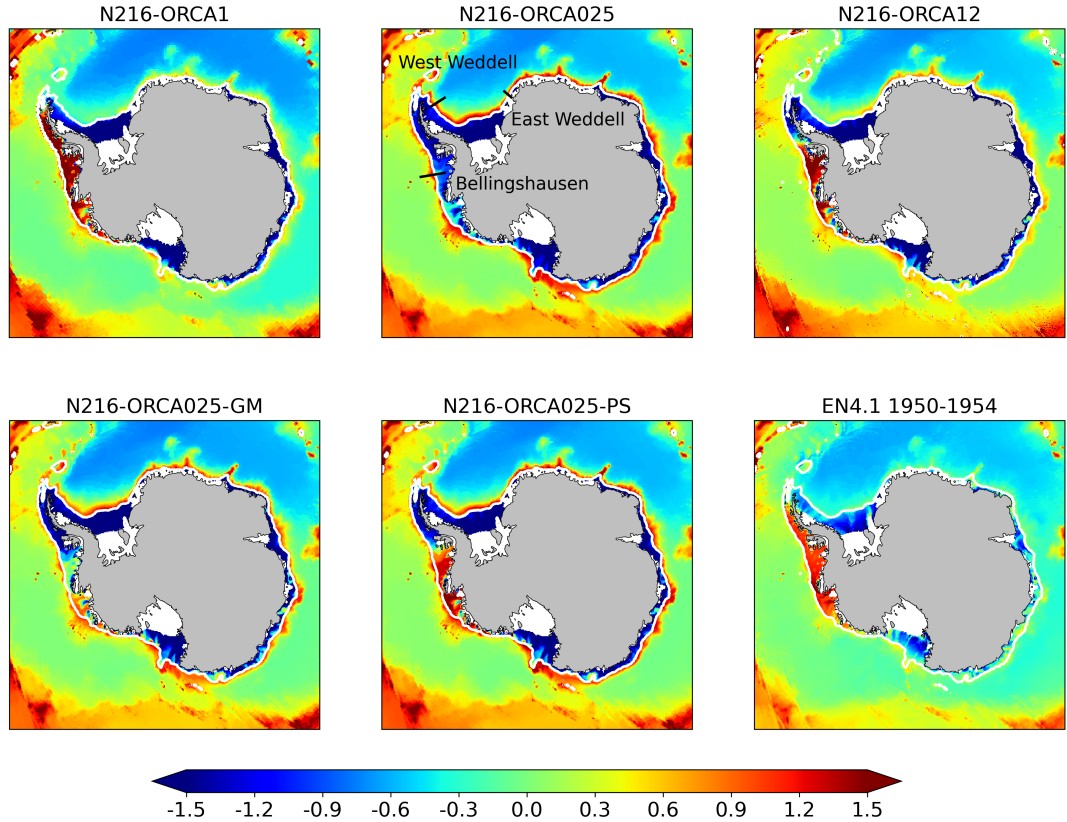

**Figure 5.** Bottom potential temperature ($^oC$), time mean for the third decade of the spin up. The experiment labels are defined in Table 2. The 1000m depth contour is shown in white. Bottom temperatures from a 1950-1954 climatology of the EN4.1 reanalysis (Good et al., 2013) are shown in the bottom right figure. The top middle figure shows the approximate locations of the sections described in Thompson et al. (2018) and plotted in Figure 8.

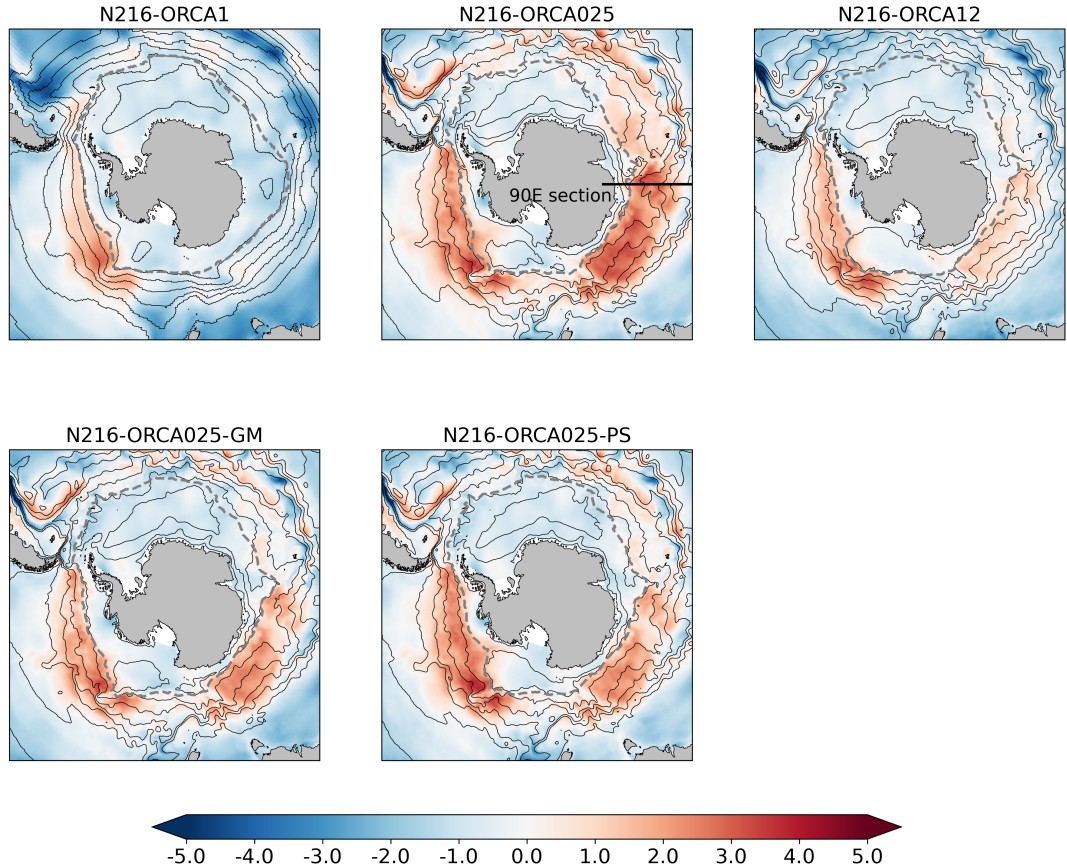

**Figure 6.** SST anomalies against ESA-CCI-SST (colours). Also shown are mean SSH (thin black lines) to show the positions of the gyres and the ACC; and the mean September ice extent indicated by the contour of $15\%$ ice concentration (thick dashed grey line). The experiment labels are defined in Table 2. Time means for the third decade of the spin up. The line in the top middle plot shows the location of the $90°$E section displayed in Figure 9.

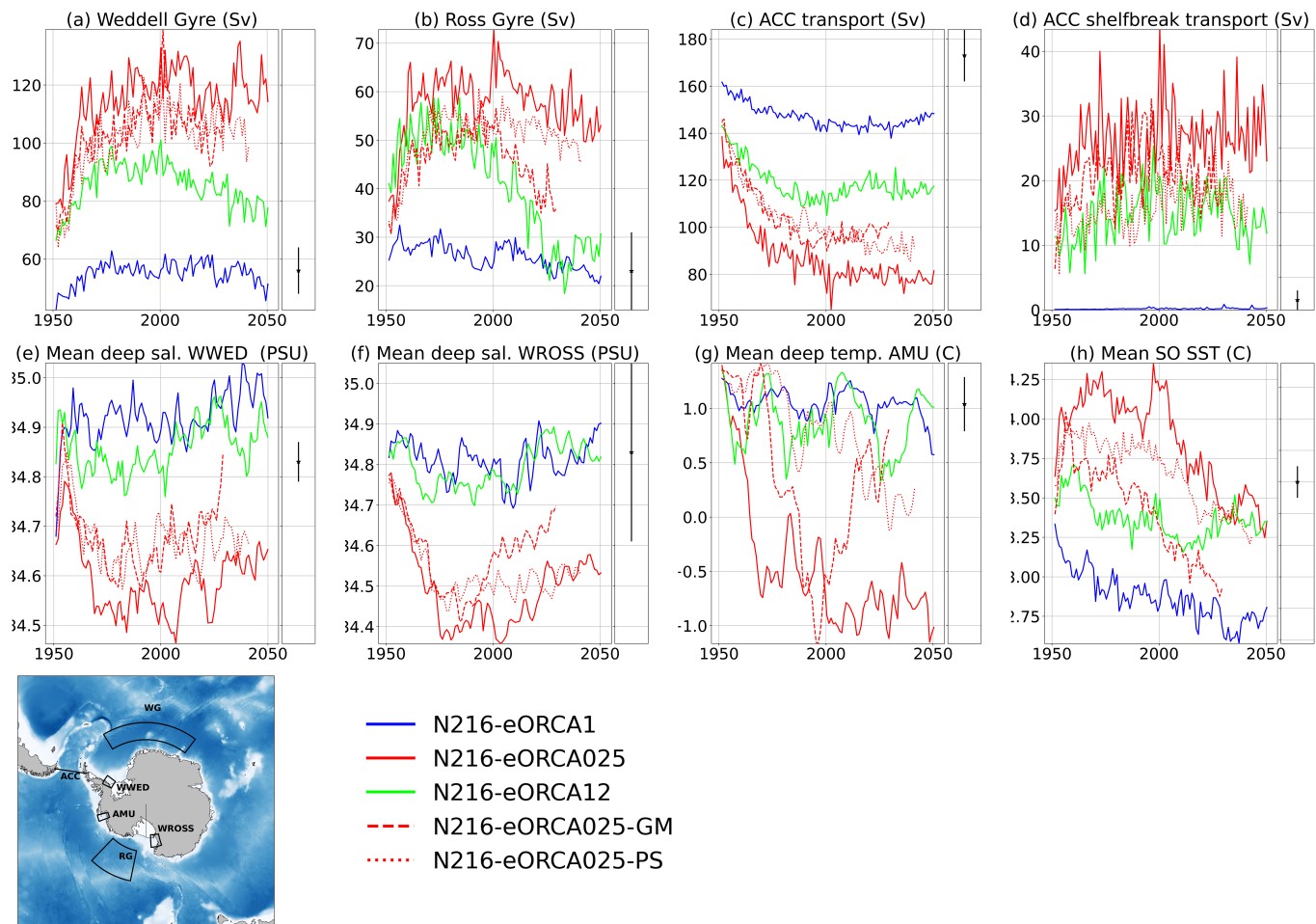

**Figure 7.** Timeseries of Southern Ocean metrics for the first 100 years of the HighResMIP integrations. Experiment labels are defined in Table 2. The plotted quantities are annual means apart from the WWED metric where summertime (DJF) means are used to better match the available observations (see Appendix A). Observational estimates and uncertainties are plotted as the black dots and lines to the right of the timeseries plots. From top left: a) The transport of the Weddell gyre plus ASC as indicated by the maximum streamfunction in the WG box, compared to the estimate of Klatt et al. (2005); b) The transport of the Ross gyre plus ASC as indicated by the maximum streamfunction in the RG box, compared to the estimate of Dotto et al. (2018); c) The net transport in the Drake Passage compared to the estimate of Donohue et al. (2016); d) The total westward transport in the Drake Passage south of 62S compared to the estimate of Meijers et al. (2016) e) The salinity below 400m spatially averaged over the WWED box in the western Weddell Sea; f) The salinity below 400m averaged over the WROSS box in the western Ross Sea; g) The temperature below 400m averaged over the AMU box in the Amundsen Sea; The deep temperatures and salinities are compared against time and spatial means of profiles from the EN4.2.2.g10 dataset (Good et al., 2013) as described in Appendix A; h) The SST averaged between 45S and 70S compared to the 1950-1954 climatology of the EN4.1.1.g10 analysis (Good et al., 2013) averaged over the same region.

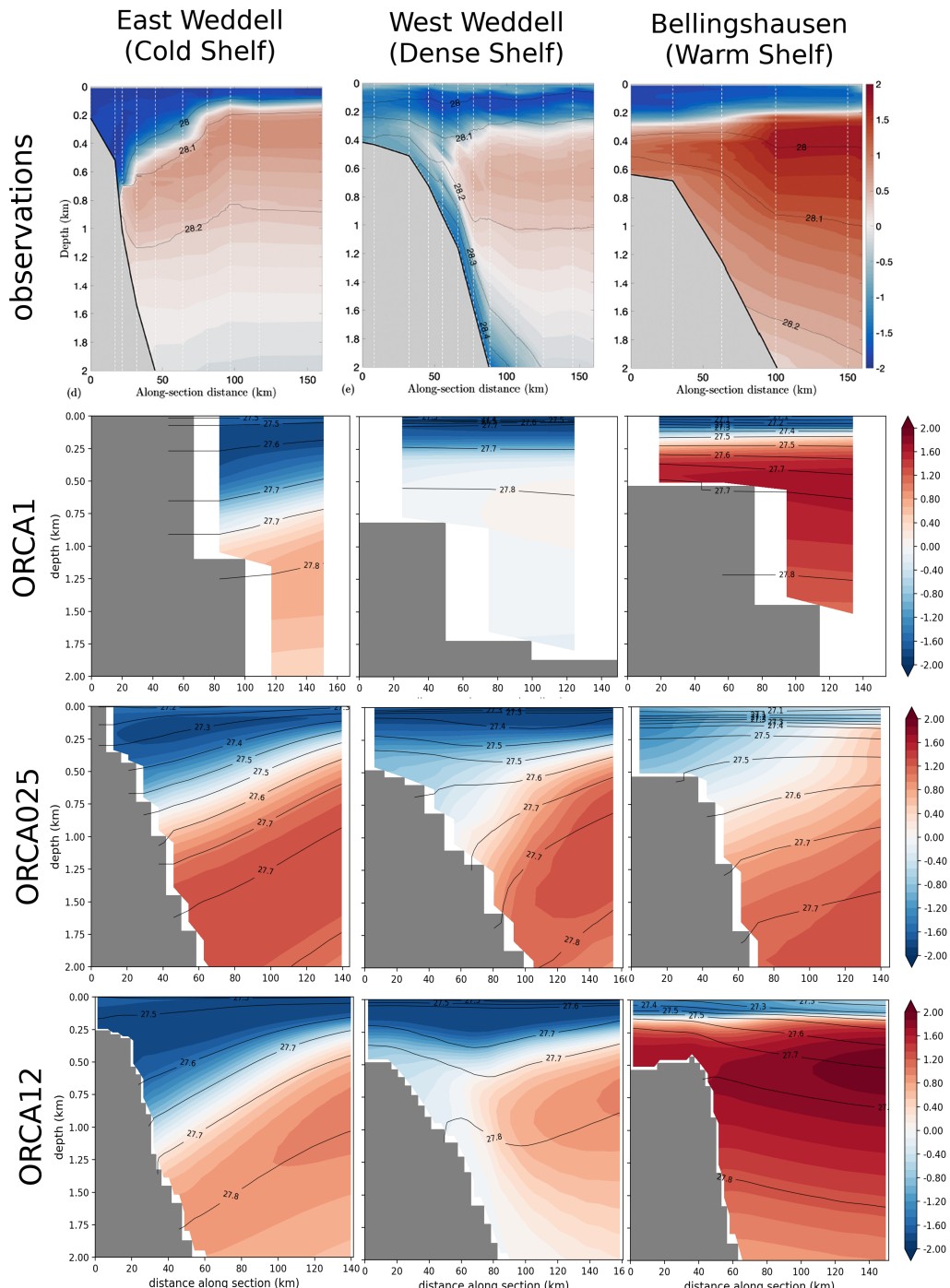

**Figure 8.** Sections of potential temperature (colours) and potential density ($\sigma_0$ - lines) across the shelf slope for three locations around Antarctica comparing the observational sections plotted in Thompson et al. (2018) with the model data. Plots of model data show time-mean from the third decade of the integration. Top row: observations - figures adapted from Thompson et al. (2018); subsequent rows: model data. The approximate locations of the three sections are shown in Figures 4 and 5

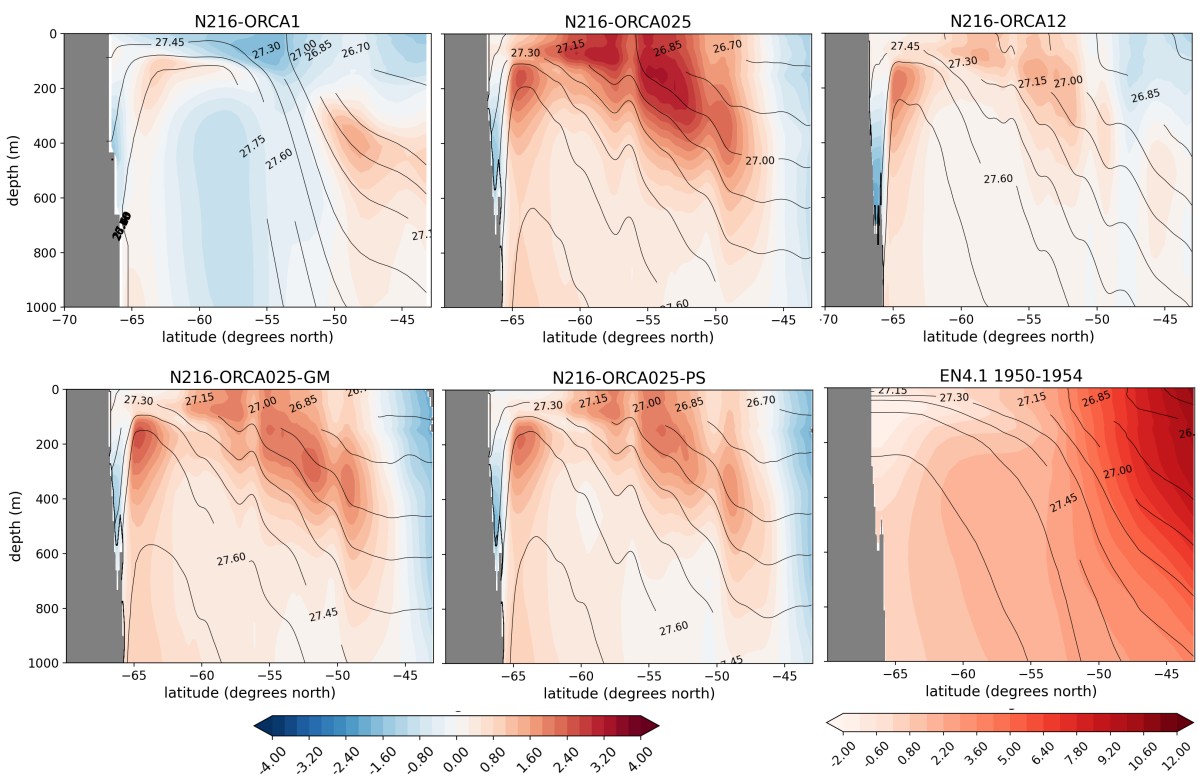

**Figure 9.** Section at 90°E (marked in Figure 6) showing temperature anomalies against EN4.1 (colours) and potential density ($\sigma_0$ - lines) for the various integrations, and potential temperature and potential density for the EN4.1 climatology. Model fields are time means for the third decade of the integration.

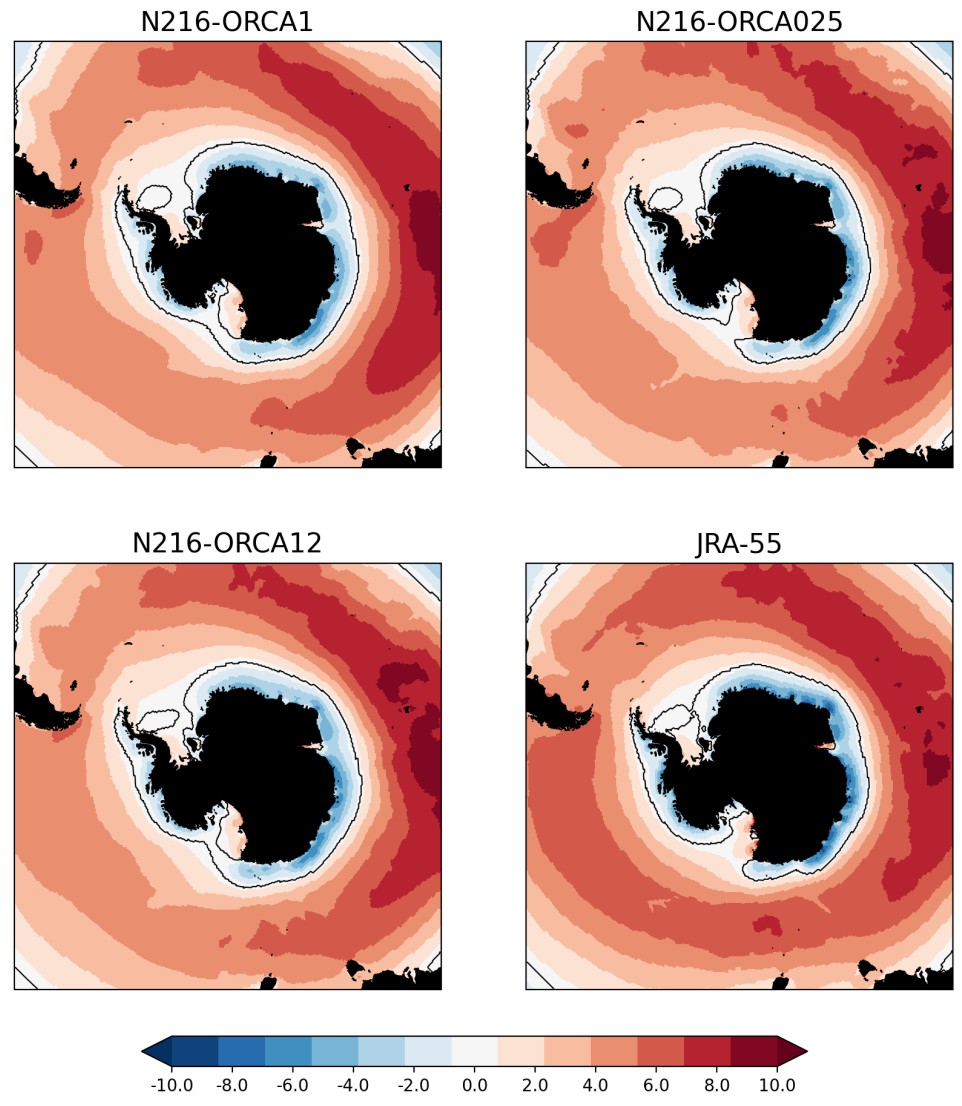

**Figure 10.** Zonal winds (m/s) at 10m for the three control integrations and the JRA-55 reanalysis (Kobayashi et al., 2015). The black lines show the contour of zero zonal wind component. The fields are 10-year means for the third decade for the model integrations and means over an equivalent period (1970-1979) for JRA-55.

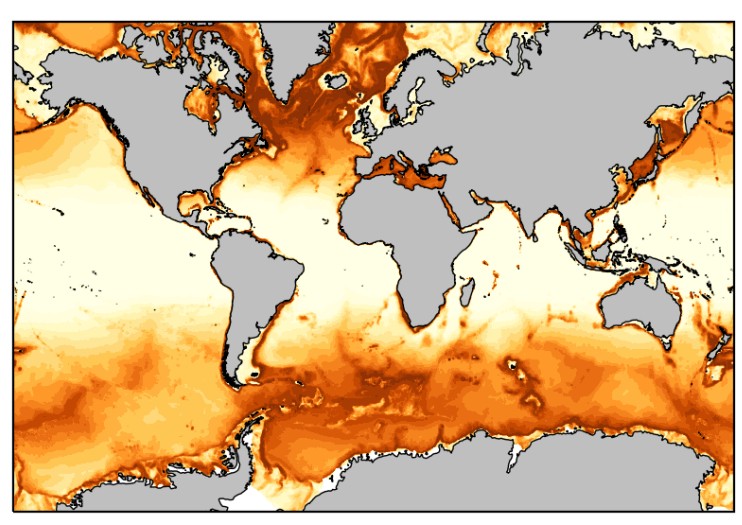

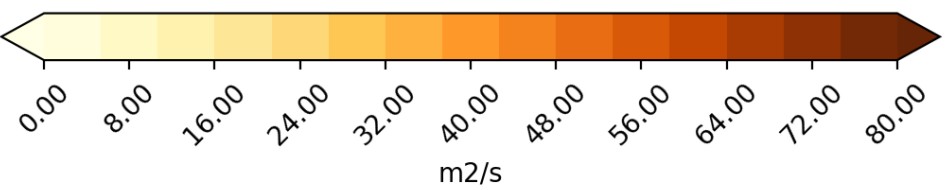

m2/s

**Figure 11.** 1-year mean Gent-McWilliams coefficient with Rossby-radius-dependent cap for the eddy-permitting model.

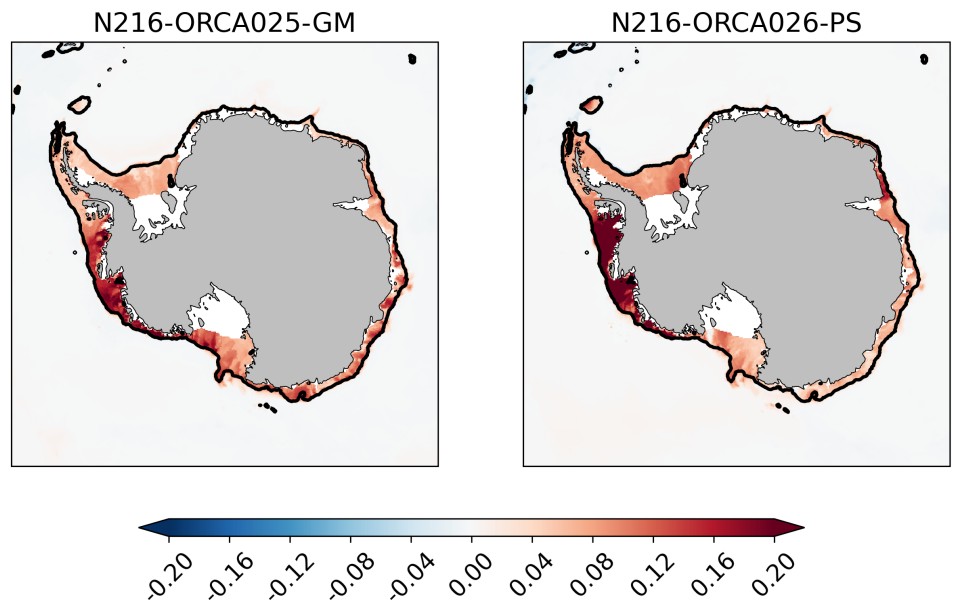

**Figure A1.** Bottom salinity (psu): differences between the N216-ORCA025 sensitivity experiments and the control for the 10-year mean fields shown in Figure 4.

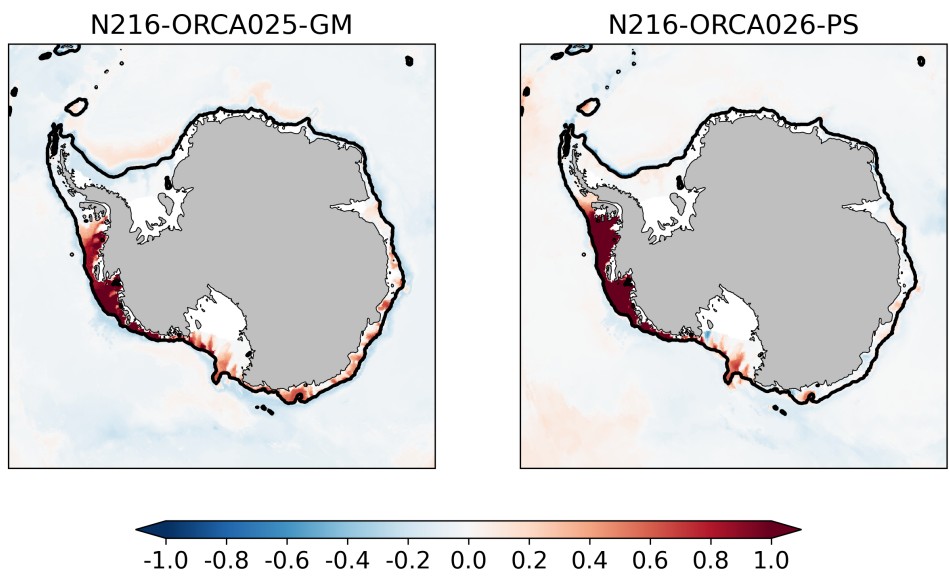

**Figure A2.** Bottom temperature (°C): differences between the N216-ORCA025 sensitivity experiments and the control for the 10-year mean fields shown in Figure 5.

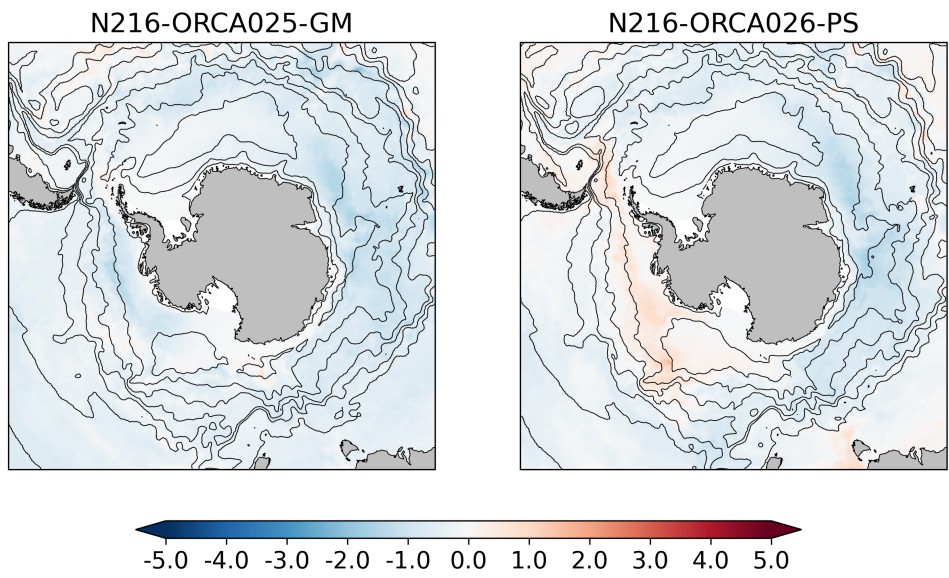

**Figure A3.** SST (degC): differences between the N216-ORCA025 sensitivity experiments and the control for the 10-year mean fields shown in Figure 6. Also shown are line contours of mean SSH to help to orientate the differences with respect to the gyres.

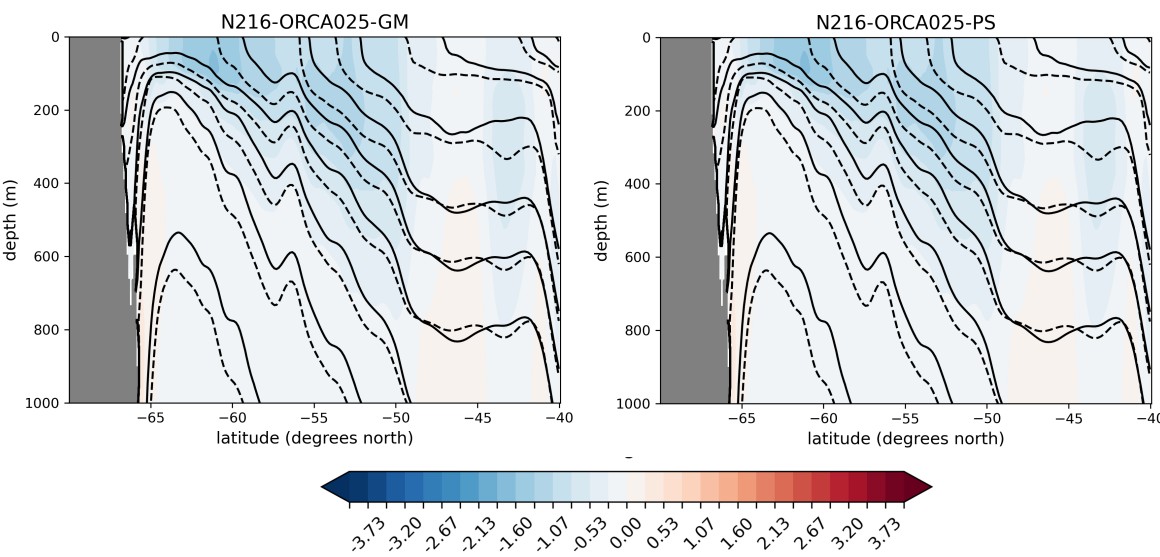

**Figure A4.** Section along 90°E showing temperature differences (colours) between the N215-ORCA025 sensitivity experiments and the control for the 10-year mean fields shown in Figure 9. Also shown are line contours of potential density ($\sigma_0$): dashed lines control; solid lines sensitivity experiment.