# Peer review of "Resolution dependence of interlinked Southern Ocean biases in global coupled HadGEM3 models"

_EGUsphere, 2024_

## Referee Comment (RC1)

Review of: *Resolution dependence of interlinked Southern Ocean biases in global coupled HadGEM3 models*.

The authors present an analysis of the Southern Ocean state during the spin-up stage for three versions of the HadGEM3 model with two versions in which Gent-McWilliams and bathymetric drag are adjusted. The three versions of HadGEM3 include models which are "non-eddying" (nominal 1-degree), "eddy-permitting" (nominal 0.25-degree), and "eddy-rich" (nominal 1/12-degree). The "eddy-permitting" version is in the grey area of resolutions where no GM is utilized, but eddies are not completely resolved at high latitudes. Two additional versions of the "eddy-permitting" model are used to identify the role of mesoscale mixing parameterized by GM and bottom drag in simulating the Southern Ocean and Antarctic margin state. The authors find introducing baroclinic stability via GM or changing bottom drag results in a generally improved simulation for the eddy-permitting model. However, even with this bias reduction, for many of the metrics analyzed here, the Southern Ocean remains far from observed (very weak ACC, too strong ASC, too strong gyres, too fresh shelf).

The manuscript is well written, and the results are useful to the community. My main concerns are outline below followed by more specific comments.

**General comments:**
- If the authors wish to make statements regarding the ASC strength, this needs to be separated from the Gyre. The ASC strength should be assessed at particular locations along the slope or via the along-slope flow. See my notes below on this topic.
- Language of "too-strong", "too-weak". The comparisons are being made against a 1950-54 climatology based on EN4 ......  so, this language needs to be careful. For a 1950-54 climatology, observations over this period and particularly in the Southern Ocean are going to be extremely sparse. Thus, an assessment of realism using a 1950-54 climatology as "truth" is not accurate.
- 150 years is relatively short, have these simulations been ran out longer and the ASC does not weaken to lower values in the eddy-permitting model?
- Complementary analysis of DSW formation / export and AABW export would provide additional insight to the processes discussed in the manuscript and particularly the shelf hydrography. A disconnect between DSW formation and AABW export might provide some insight into magnitude of interior / spurious mixing.

**Overall:** I recognize this is a non-trivial problem and I am still puzzled by it. Even with the added GM and additional drag, the ACC is still exceptionally weak due mostly to a huge component of westward flow coming from the overly-large Weddell Gyre & ASC. Does the basal melt introduced at depth play any role in this? (thinking compared to models that do NOT account for this). Could the buoyancy introduced in this manner and the lack of resolved eddies play any role?

**L16**: "Antarctic Overturning Circulation" → I think stating "between the near surface and deep ocean via the formation of Dense Shelf Water (DSW), Antarctic Bottom Water (AABW), and mid-depth ocean via mode and intermediate water formation" would work better here. If you are linking to the Southern Ocean being critical to the climate system --- also mentioning the mode & int water formation is important as this is where all the heat and carbon is going.

**L23:** Capitalize C D and W in "Circumpolar Deep Water".

**L24:** Also, there are a lot of warm biases in 1-degree CMIP-class models (Beadling et al., 2020), not sure there is a definitive link to high resolution models being warmer? For example, in some high-res simulations we actually see warm biases improve with finer nominal grid spacing.

**L25:** It would be useful to cite Hallberg 2013 here which shows this limitation very well.

Hallberg, R. (2013). Using a resolution function to regulate parameterizations of oceanic mesoscale eddy effects. Ocean Modelling, 72, 92–103. https://doi.org/10.1016/j.ocemod.2013.08.007.

**L50-53:** This paragraph is probably not necessary.

**L56-57:** These two sentences appear as separate from the paragraph below. I suggest combining these into the paragraph below.

**L60:** "and partial cells (Barnier et al., 2006; Adcroft et al., 1997) allowed next to topography." → "and partial cells at the ocean bottom to better represent bathymetry (Barnier et al., 2006; Adcroft et al., 1997)."

**L67:** "Cavities under ice shelves are closed and the output of basal melt water at the ice shelf front parametrised as described in Mathiot et al. (2017)." → This is interesting, so this model represents "ice shelf melt"? Is this just based on some threshold of solid precip over the Antarctic continent? Could you elaborate on this? I assume this does not imply there are realistic melt rates?

**L83-84:** I suggest to briefly elaborate on the term "model drift" here for readers that are not familiar.

**L84-85:** "However the early spin up of the model can be useful in diagnosing model biases, since at this stage the model has not drifted too far from initial conditions." ← I am not sure I agree with this statement, assuming the ocean is starting from a present-day climatology (say WOA13 or WOA18) and a pre-industrial atmosphere, this early stage is an unrealistic climate state and an assessment of realism (i.e., biases relative to observed) is better made once the model has been able to achieve its own equilibrium (or better yet,

reached that equilibrium and forced with observed climate forcings; i.e., the historical simulation). I tend to think of the spin-up stage as the adjustment stage that we don't want to consider when doing assessments against observations. I suggest rewording this or expanding on your reasoning here.

**L93-101:** As you note, the gyres and ASC transports merge into one another particularly in the Weddell, so it is hard to discern these from one another in the current figures. I would suggest an additional plot of the upper 1000 m speed (or upper 500 m speed) to see the differences in the strength and location of the ASC.

**L102:** "net eastward transport" → this wording is confusing, there is eastward and westward flow through Drake Passage, should this just say "net" to avoid confusion?

**L103-104:** It is worth mentioning that this is exceptionally weak even compared to earlier / other estimates (Cunningham et al., 2003; Griesel et al., 2012; Meijers et al., 2012; Koenig et al., 2014; Firing et al., 2011; Xu et al., 2020)

**L108:** "These counterflows significantly reduce the net eastward transport." I would remove the word eastward after net and just say "these westward counterflows significantly reduce the total net transport".

**L109-111:** It is worth mentioning that Xu et al., (2020) also shows net westward flow at depth -→ this is why the authors argue that Donohue et al., (2016) overestimated the net transport through Drake Passage. Although they are referring to bottom recirculations --- different from what is shown here.

**Figure 3 caption:** Why is the 1950-54 climatology used here for comparison? Because it is close to the initial conditions?

**L120:** "partially eroded **to** the east of this region in the eddy-rich model" -→ swap "in the east" to "to the east". Also, this is likely due to the delivery of cold, fresh Weddell Sea water to the WAP.

**L122-128:** How different are the sea ice edge locations? It would be helpful to add these to the plots of Figure 5 for reference of where these anomalies are.

**L131-133:** Given that the gyres and ASC are governed by different dynamics, I strongly suggest breaking this down into an assessment of gyre strength and the ASC separately. The ASC could be computed as the total transport from the slope southwards at specific transects (this will also include the Antarctic coastal current too so this will not be perfect) (as in Moorman et al., 2020; Beadling et al., 2022) or as the total along-slope flow as in Huneke et al., (2022).

https://journals.ametsoc.org/view/journals/phoc/52/3/JPO-D-21-0143.1.xml

https://agupubs.onlinelibrary.wiley.com/doi/10.1029/2021JC017608
https://journals.ametsoc.org/view/journals/clim/33/15/jcliD190846.xml

**L133:** remove "eastward" before "net" due to comments earlier.

**L131-135:** These series of sentences sound a bit choppy as they are currently written with all starting with "We", "We", "We".

**L134:** Remove "deep" before "fields below 400 m" as "fields below 400 m" imply they are deep.

**L143:** "and comparing to a similar average performed on the **1950-1954** climatology of the EN4.1.1.g10 analysis dataset". I might be missing something, but why is this time period used for comparison? Observations would be very sparse for this time period and particularly so in the Southern Ocean.

**L157-158:** Yes, this could indicate and issue with CDW cross-shelf intrusions, but this could *also* be due to the westward transport of cold, fresh Weddell Sea water around the WAP mixing with CDW (making it colder & fresher). The maps of ocean velocities support this connection. This has been found in other simulations when the ASC accelerates (Beadling et al., 2022) and this mechanism has been documented as well by Morrison et al., (2023) "Weddell Sea Controls of Ocean Temperature Variability on the Western Antarctic Peninsula"

https://agupubs.onlinelibrary.wiley.com/doi/full/10.1029/2023GL103018
https://agupubs.onlinelibrary.wiley.com/doi/10.1029/2021JC017608

You mention this below in lines 168 – 169 but it should be mentioned here or this discussion combined.

**L163-165:** This would be shown nicely with a surface water mass transformation analysis (sWMT). This is also consistent with Tesdal et al., (2023) which showed that when the ASC accelerates, DSW reduces as the shelf becomes more buoyant.
https://agupubs.onlinelibrary.wiley.com/doi/full/10.1029/2022JC019105.

**L180:** Add "ocean" before circulation.

**L183:** Should you add "and associated treatment of mesoscale eddies" after "ocean resolution", to be clear this is not just due to horizontal grid spacing *alone?*

**L201:** remove "eastward" when referring to the net transport.

**Figures 1,3,4,5,8** → It would make comparisons easier for the reader to add two additional panels to these plots of the N216-ORCA025-GM **MINUS** N216-ORCA025 and N216-

ORCA025-PS **MINUS** N216-ORCA025. It is hard as of now to discern some of the differences.

**Figure 2 & L200-203.** It looks like GM really impacts the westward along-slope flow (ASC) through the passage while the rest appears unchanged. PS appears to reduce flow everywhere (even the eastward flow in the Subantarctic Front), however reduces the eastward components more ... which is why the net increases. It is hard to see visually what component of the along-slope flow is decreasing --- **is this mostly coming from the bottom flow or surface intensified flow?**

**L204:** "The fresh biases on the shelves are reduced, with some recovery of the HSSW in the western Weddell Sea and western Ross Sea (Figure 3)" -→ The shelves in the South Indian sector appear relatively unchanged too.

**Figure 6:** The lines for N216-eORCA025-GM and N216-eORCA025-PS are very hard to discern. Can you make one have circle markers? The dashed and the dashed-dotted are very hard to distinguish.

**L205:** "The timeseries show that again, Gent-McWilliams appears to have a stronger impact than partial slip." → This sentence is referring to shelf salinity, yet this is only true for the Ross. The PS West Antarctic shelve Amundsen / Bellinghausen) looks better for the PS (Figure 3) (This is ALSO true for shelf temperature as you mention below, so this would just require rewording). The timeseries for the Weddell salinity looks similar between the two.

**All figures:** Increase size of text on color bars / axes, some of these are hard to see.

**Figure 7:**
- The top figures from Thompson et al. (2018) have a y-axis in km, but the rest are in m. This should be made to be consistent across the panels. Text on axes are also hard to read. The titles on the top and bottom also look very large compared to the other labels in the other figures in the manuscript.
- **Figure 7:** I assume that the grey shaded regions are not the models true bathymetry? The blocky-nature makes it appear that the models do not account for partial cells. I assume in reality this is more smooth?
- **Figure 7:** The Thompson et al., figures are conservative temperature, not potential temperature. These should be consistent between the observation panels and the model output panels.

**L220:** "The properties of the shelf water are controlled partly by local surface fluxes and partly by the exchange of water with the open ocean across the shelf break" → "The properties of the shelf water are controlled partly by local surface fluxes and associated water mass transformations, and partly by the exchange of water with the open ocean across the shelf break"

**L223-226:** This paragraph seems short for a stand-alone paragraph, consider merging with the one below.

**L229:** This statement needs a citation: "these incursions are likely to only happen occasionally as tidal or eddy driven fluctuations of the front position onto the shelf."

**L237:** Cite Thompson et al., (2018) after this statement: "the observed structure is more complex, with a V-shaped pattern of isopycnals associated with the incursion of CDW onto the shelf and its transformation and export as Dense Shelf Water (DSW)".

**L255 (and throughout):** Degree sign missing between number and letter for longitude (this is also true for latitudes in the text).

**L257:** Add comma after "in general".

**L268-270:** Paragraph is short – suggest combining.

**L276:** This is consistent with the feedback to meltwater Bronselaer et al., (2018) suggested i.e., slumping of isopycnals resulting in more heat delivery to shelf --- just pointing it out, but perhaps not necessary to discuss.

**L281:** Add "ocean" before flow.

**L284:** Should this be "wind stress curl", not "wind curl"?

**L285:** What does "medium resolution" imply in km?

---

## Referee Comment (RC2)

Storkey and colleagues run the climate model HadGEM3 using different nominal resolutions (1°, 1/4°, 1/12°) of the ocean component NEMO. They find that the model version with 1/4° resolution (referred to as eddy-permitting) has stronger Southern Ocean biases than the other versions. They hence proceed to attempting to reduce those biases by 1) adding an eddy-parameterisation to compensate for the resolution being insufficient to resolve all eddies, and 2) increasing bathymetric drag along the lateral boundaries. Both approaches act to dampen the overactive circulation in subpolar gyres and the Antarctic Slope Current (ASC), which leads to improvement in the biases. The authors link near-shelf biases to overly steep isopycnals and a too-strong ASC, preventing Circumpolar Deep Water (CDW) from reaching the shelf. They also show that the weak Antarctic Circumpolar Current (ACC) transport and near-surface warm biases in the open ocean can be linked to too flat isopycnals across the ACC.

The manuscript is generally well-written, with illustrative figures and a comprehensive discussion of the findings, though there is still room for some improvement. This work is of interest to other modelling groups running simulations at eddy-permitting ocean resolution, in particular those using NEMO configurations, and can help contribute to the understanding of origins of Southern Ocean biases in models, which is a common issue that is not easily resolved and occurs across a broad range of resolutions. This work does not solve all the issues in the eddy-permitting simulations (e.g., the ACC is still weak), and simulations are quite short, which I recognise as likely due to limited computational resources, but it is an insightful step on the way. Below, I outline my recommended revisions before publication of this manuscript.

**General comments**

- Upon reading the manuscript, I was immediately puzzled by the counterintuitive result that resolutions that are only partly eddy-resolving end up with too-flat isopycnals in the ACC (flatter than the resolution with parameterised eddies), despite supposedly not achieving the full extent of eddy compensation. To me, overly flat isopycnals in the ACC would suggest that there is likely an overcompensation by the eddy field to counterbalance the northward Ekman transport achieved by the prevailing wind field. I was asking myself how the authors explain the over-flattening of the isopycnals in the higher resolution model simulations, particularly when adding the scale-aware G&M in ORCA025 (i.e. adding more of an eddy effect) leads to steeper isopycnals across the ACC. A tentative explanation is later given in Conclusions, and noted as a topic for future work, but for the reader, it may be beneficial to acknowledge this counterintuitive characteristic of the simulation much earlier. I would prefer to see it discussed a bit more extensively, including a mention of how it could potentially be explained, already in the Discussion section, and then summarized in the Conclusions. Noting here that, in e.g., the Weddell Gyre, isopycnals are too steep in the eddy-permitting resolution, and adding G&M leads to flattening and a less active gyre, as expected.
- I can see how adding the scale-aware G&M and/or changing the slip condition would also be beneficial in the eddy-rich resolution. Has this been tested at all by the authors (I acknowledge that this is not the focus of the current manuscript)? If so, which of the two changes is expected to be most beneficial to alleviate biases at that resolution? A brief discussion on this could make the manuscript even more valuable to the broader ocean modelling community.

- In line with the previous review comment, I also note that the use of the EN4 1950-1954 climatology as an observational reference thus calls for some caution in phrasing.

**Specific comments**

Abstract

L. 1-2: "eddy-permitting ocean resolution" → "eddy-permitting ocean resolution without additional eddy parameterisation"

L. 8: "unresolved eddy processes or the representation of bathymetric drag" → and/or, since both can be (and are) causing issues at the same time.

L. 9: Already here in the abstract, the authors mention the shallower isopycnal slopes of the eddy-permitting resolution, and it immediately caught my attention as being counterintuitive. Hence, I would have liked to see this acknowledged earlier in the paper itself.

L. 30-32: It would be appropriate to also mention the emergence of models with unstructured grid configurations, as they are an approach to overcoming the issue with affording high-enough resolution to resolve high-latitude eddies in global models, e.g. *FESOM* (Wang et al., 2013, https://doi.org/10.5194/gmdd-6-3893-2013 ; Scholz et al., 2019, https://doi.org/10.5194/gmd-12-4875-2019 ), *ICON* (Jungclaus et al., 2022, https://doi.org/10.1029/2021MS002813;Korn et al., 2022, https://doi.org/10.1029/2021MS002952 )

L. 61-62: "A free-slip boundary condition […]" - It would be valuable to also mention what viscosity scheme is used, and potentially also the parameter settings since they likely differ between resolutions, as these also have the potential to affect the flow and thus the biases.

L. 64-65.: "Diffusion of tracers along isopycnal surfaces, parameterising eddy mixing […]" - Is there any regional reduction of the parameterised eddy mixing around the equator where eddies are fully resolved, in particular in the higher resolutions?

L. 75: "with the N216 atmosphere" - Are the results consistent or at least similar if one of the other atmospheric resolutions are chosen? This would indicate that the results are more widely applicable to other coupled models, regardless of what atmospheric setup they use.

Page 3, footnote 1: Was the eddy-permitting model ever tested with no-slip? It would be useful to motivate the choice of partial-slip over no-slip in this case, and discuss how choosing a no-slip condition might have impacted the overly strong ASC. Useful references may be Penduff et al., 2007, https://doi.org/10.5194/os-3-509-2007 ; Deremble et al., 2011, https://doi.org/10.1016/j.ocemod.2011.05.002; Nasser et al, 2023, https://doi.org/10.1029/2022MS003594

L. 93-95: "As well as having more active gyres, the higher resolution models also have a stronger ASC […]" - In the next paragraph, you mention the consequences of this feature in the Drake Passage, but only after you discuss the over-flattened isopycnals. This makes this

part of the text feel somewhat fractured. Maybe mention the Drake Passage briefly already here, or rearrange the next paragraph to discuss the ASC behaviour before the overly flat isopycnal slopes.

L. 98-99: Klatt et al. (2005) is one of few observationally-based estimates of the Weddell Gyre strength published in recent decades. It is, however, not the only one. Reeve et al. (2019, https://doi.org/10.1016/j.pocean.2019.04.006) estimate it to 32 +/- 5 Sv based on ARGO data. Older observational estimates by Farbach et al. (1991) and Yaremchuk et al. (1998, https://doi.org/10.1007/s00585-998-1024-7) are also lower c.f. Klatt et al. Meanwhile, the transport in models ranges between at least 10-80 Sv (see e.g., Neme et al., 2021, https://doi.org/10.1029/2021JC017662; Wang, 2013, https://doi.org/10.1002/jgrc.20111)

L. 105-106: "The weaker ACC transport in the higher resolution models is associated with a flattening of the time-mean isopycnal slopes in the Drake Passage" - Given that particularly the ORCA025 resolution is not fully eddy-resolving in the ACC region, it is somewhat counterintuitive that there is an over-flattening of the isopycnals, which suggests too much eddy compensation. From reading the text here, it makes one wonder how that can be. As mentioned in the general comments, this counterintuitive behaviour is mentioned by the authors later in the manuscript, but should be acknowledged sooner. However, looking at the figures, it looks like the isopycnals are very steep (steeper than ORCA1) in the northern part of DP, where the core of the ACC is, and then flattened in the centre, where there is weaker(ORCA12)/counter(ORCA025) flow. In ORCA12, steeper isopycnals than in ORCA1 are also observed between -63 and -62 degN. These details should be mentioned, as they might help elucidate what is actually happening to the ACC transport.

L. 107: "counterflowing currents […] associated with the Shackelton fracture zone" – The authors mention that the modelled counterflow along the southern shelf break is unrealistic c.f. Meijers et al. (2016), but they give no indication of whether counter flows at the Shackelton fracture zone correspond to observations or not.

L. 122-128: On resolution-dependent temperature biases - What are the differences in global mean surface temperature in these simulations, and compared to the observational dataset? As some of the biases in SST can stem from the atmospheric model, or results of the difference in resolution in other regions, it might be useful to make a supplementary figure where the biases are normalised to the observational global mean surface temperature.

L. 132: "observational estimates of Klatt et al. (2005)" – as this is not the only observational estimate of the Weddell Gyre strength (see L. 98-99), this may need to be modified, or at least motivated why this particular estimate is used for comparison.

L. 165: "in the eddy-permitting model […] deep water formation has been suppressed" - Does this lead to (more) unrealistic open-ocean convection than in the other two model versions?

L. 168-169: Cold, fresh water advecting around the Antarctic peninsula in the eddy-rich model suggests the ASC and is too strong through the Drake Passage also in this model version.

L. 185-189: On introduction of the scale-aware G&M: What are the implications of running a resolution that to some degree allows eddy formation, but then also parametrising eddies on top of it using G&M, which can lead to "smoothing out" of the actual eddies? This should be clarified in the text.

L. 190-192: On the introduction of the partial-slip condition – As mentioned above, it would be useful to motivate why the choice was to go with partial-slip and not no-slip for this resolution. Also, based on the description of how partial slip affects the biases in ORCA025, it seems that introducing it south of 50S in ORCA12 could potentially fix the remaining biases with counter flows in the Drake Passage, and thus with cold waters being advected around the Antarctic Peninsula in this resolution as well.

L. 223-224: "the Fresh Shelf, the Dense Shelf and the Warm Shelf. In Figure 7" - It would help guide the reader's eye if it were also indicated in the figure and/or the figure caption what column exemplifies which one of these three regimes.

L. 238-239: About the V-shaped pattern of isopycnals in the higher-resolution models - I struggle to find this V-shape in the drawn isopycnals in the eddy-permitting resolution. If so, I can only see it in the two isopycnals labelled 27.5 (this might be a mistake in the labelling, as it occurs twice).

L. 262-265: The EN4. 1 climatology does not show the same steep ispoycnal slopes near the continent as the model in this region but, as mentioned, the observations included in the climatology are sparse. It could be useful to cite other data (not included in the EN4.1 climatology) that give indications about the isopycnal structure in the area even if those are from a later time period.

L. 310-320: On open-ocean polynyas - Are these events stronger/more frequent in the eddy-permitting c.f. the eddy-rich resolution, given that the latter appears to have some more capability of forming dense water on the shelves? (see also L. 165)

L. 332-336: About biases along the continental slope/shelf – Here, it would be helpful to clarify which factor is the more important in reducing these biases: adding G&M or introducing the partial-slip condition.

Figure 1 (caption): See comments to L. 98-99 and L. 132 regarding Klatt et al. (2005)

Figure 5: mean SST-lines - It is unclear to me which mean SST this refers to, and as the lines are completely unlabelled, there is no indication of what temperatures the different lines represent.

Figure 9: In this figure, it might be more illustrative to show the model results as anomalies from the reanalysis data. In the other figures, observational datasets are shown in the bottom-right subpanel. It would be helpful to keep the same structure throughout the manuscript.

Figures overall: It could be helpful with supplementary figures showing the differences between the standard N216-ORCA025 and the other two ORCA025 versions (GS and PM) as anomalies from the standard (GS-standard, and PM-standard)

---

## Author Comment (AC1)

*Resolution dependence of interlinked Southern Ocean biases in global coupled HadGEM3 models.*

**Authors' response to reviewers**

We would like to thank both reviewers for taking the time to provide detailed and constructive reviews.

Here we respond to the general comments of both reviewers first and take the more detailed comments later. Where we have not responded explicitly to minor comments and corrections we would propose to follow the reviewer's suggestion.

**General comments**

1. Separation of the recirculating gyre transport and ASC transport. Reviewer 1 makes the point that the dynamics of the gyre and the slope current are distinct and suggests that we provide figures for the ASC transport separately by calculating the transport over the continental slope. In the region of the gyres it is difficult to separate the ASC from the gyre, since the recirculating gyre transport impinges on the continental slope and models at these resolutions do not form very distinct jets – see for example the plots below showing cross sections of currents and depth-integrated transport densities from the MM model in the Weddell gyre.

[Figure]

   In order to give some indication of the circumpolar flow distinct from the recirculating flow we propose to follow the reviewer's suggestion of providing plots of the depth-mean currents in the top 500m to complement the plots of the streamfunction in Fig 1. We will also provide an extra timeseries plot in Fig 6 showing the timeseries of the counterflow at the southern boundary in the Drake Passage (as defined in Fig 2) compared to the observations of Meijers et al (2016).

2. Use of EN4 1950-1954 climatology for assessment. Both reviewers make the point that observations in the Southern Ocean for this period are extremely sparse, and therefore advise caution in the conclusions drawn by comparing the model with the EN4 climatology for this period. The EN4 analysis uses a climatology for the period 1970-2000 as a background climatology to which the solution relaxes in the absence of observations (Good et al, 2013). Given the lack of observations in the Southern Ocean, the 1950-1954 climatology is likely to be very close to this background climatology. For a long-term climatology to compare the model to this is the best we can do, but we will elaborate on this point in the revised text.

3. Length of integration and experimental design. Both reviewers point out that the integrations are short in the context of climate studies, and reviewer 1 questions the statement that *"the early spin up of the model can be useful in diagnosing model biases, since at this stage the model has not drifted too far from initial conditions"* because the initial state of the ocean is out of balance with the greenhouse gas forcing being used and there will be an adjustment associated with this. These experiments follow the HighResMIP protocol (Haarsma et al., 2016), in particular the *spinup-1950* and *control-1950* experiments, in which the model is initialised with "1950s" initial conditions and spun up with 1950s greenhouse gas forcing. As stated by Roberts et al (2019), the nominal spin up for these integrations is very short (30 years) due to computational constraints. As noted in point 2 above the "1950s" initial conditions are likely more similar to a later state of the ocean, so there is an imbalance between initial conditions and the forcing, but the imbalance will not be as great as for a pre-industrial spin up integration. Still it is clear that the model is adjusting in the first decades of the spin up and nowhere near an equilibrated state. However from the timeseries in Fig 6 it seems that there is a fast initial adjustment over the first 2-3 decades after which the three models reach very different states, and these differences in many cases persist for multi-centennial timescales. (We have only shown the first 100 years in Fig 6 because of the length of the sensitivity experiments). In particular, the biases in the MM model appear to be very persistent, so we believe we can learn something about them by studying the initial adjustment period. We will try to make these points more clearly in the revised manuscript.

4. Dense water formation and export. Reviewer 1 suggests some analysis of dense water formation and export from the shelf and AABW export to shed light on the water mass biases on the shelf and possible spurious mixing in the interior. We agree that the models' representation of dense water formation and export could well be relevant to the formation of the biases studied in the paper. From the cross sections in Fig 7 it seems clear that the export of dense water from the

shelf in the Weddell Sea is only captured by the 1/12˚ model. We plan to do a systematic study of the dense water formation in the different models looking at the water mass transformation on the shelf, the export over the shelf break and the time evolution of the reservoir of AABW, but we would prefer to present this in another paper. We propose to include another subsection in the Discussion section, discussing the dense water formation and export and its possible links to the large-scale biases. In this section we would also discuss the possible loss of AABW due to spurious mixing (see point 5 below).

5. Explanation for isopycnal slumping in the open ocean. Reviewer 2 says that the slumping of isopycnals across the ACC in the medium resolution model is counter intuitive and that this is not addressed very prominently in the paper, with a tentative explanation only appearing in the last paragraph of the Conclusions. We agree and propose to follow her suggestion of emphasising the counter-intuitive nature of the result in the Discussion section with a discussion of the possible link to spurious mixing and the loss of AABW in the new subsection on the dense water formation and export.

6. Inclusion of partial slip and scale-aware Gent-McWilliams in 1/12˚ model. Reviewer 2 asks if we tested partial slip or scale-aware Gent-McWilliams in the 1/12˚ model. We have included the combination of partial slip and scale-aware GM in our standard 1/12˚ model configuration, but due to computational expense we only have a clean test of the impact in a forced ocean-ice configuration. In this test, the combined effect of partial slip and scale-aware GM is still positive in the sense that it increases the ACC strength and reduces the gyres, but the impact is not as great as it is for the 1/4˚ model. This might be expected because the increased resolution at 1/12˚ means that the GM coefficient will be non-zero over a smaller area than for the 1/4˚

**Detailed comments reviewer 1**

L16: "Antarctic Overturning Circulation". I think stating "between the near surface and deep ocean via the formation of Dense Shelf Water (DSW), Antarctic Bottom Water (AABW), and mid-depth ocean via mode and intermediate water formation" would work better here. If you are linking to the Southern Ocean being critical to the climate system - also mentioning the mode & int water formation is important as this is where all the heat and carbon is going. We agree that an expanded description of the overturning circulation would be good here.

L24: Also, there are a lot of warm biases in 1-degree CMIP-class models (Beadling et al., 2020), not sure there is a definitive link to high resolution models being warmer? For

example, in some high-res simulations we actually see warm biases improve with finer nominal grid spacing. Agreed – the statement about the resolution dependence may have been a Met Office centric point of view. We will remove that and just make the statement that warm biases in the Southern Ocean are a common problem in CMIP models.

L67: "Cavities under ice shelves are closed and the output of basal melt water at the ice shelf front parametrised as described in Mathiot et al. (2017)." This is interesting, so this model represents "ice shelf melt"? Is this just based on some threshold of solid precip over the Antarctic continent? Could you elaborate on this? I assume this does not imply there are realistic melt rates? The distribution of melt water input around the continent is based on Rignot et al (2013), but for these experiments the overall magnitude of melt water plus iceberg calving is scaled to equal the total precipitation over the Antarctica at each timestep, ie. An assumption that the total mass of ice over the continent is constant. We will add a sentence to this effect.

L84-85: "However the early spin up of the model can be useful in diagnosing model biases, since at this stage the model has not drifted too far from initial conditions." ß I am not sure I agree with this statement, assuming the ocean is starting from a present-day climatology (say WOA13 or WOA18) and a pre-industrial atmosphere, this early stage is an unrealistic climate state and an assessment of realism (i.e., biases relative to observed) is better made once the model has been able to achieve its own equilibrium (or better yet,reached that equilibrium and forced with observed climate forcings; i.e., the historical simulation). I tend to think of the spin-up stage as the adjustment stage that we don't want to consider when doing assessments against observations. I suggest rewording this or expanding on your reasoning here. See response under point 3 above. We will rewrite these sentences to make our approach clearer.

L93-101: As you note, the gyres and ASC transports merge into one another particularly in the Weddell, so it is hard to discern these from one another in the current figures. I would suggest an additional plot of the upper 1000 m speed (or upper 500 m speed) to see the differences in the strength and location of the ASC. L93-101: As you note, the gyres and ASC transports merge into one another particularly in the Weddell, so it is hard to discern these from one another in the current figures. I would suggest an additional plot of the upper 1000 m speed (or upper 500 m speed) to see the differences in the strength and location of the ASC. As discussed under point 1 above we will follow this suggestion and provide maps of the depth-integrated currents over the top 500m to complement the plots of the the streamfunction in Fig 1.

L102: "net eastward transport" this wording is confusing, there is eastward and westward flow through Drake Passage, should this just say "net" to avoid confusion? We will use "net transport" rather than "net eastward transport".

L103-104: It is worth mentioning that this is exceptionally weak even compared to earlier / other estimates (Cunningham et al., 2003; Griesel et al., 2012; Meijers et al., 2012; Koenig et al., 2014; Firing et al., 2011; Xu et al., 2020). We will add this point.

L109-111: It is worth mentioning that Xu et al., (2020) also shows net westward flow at depth - this is why the authors argue that Donohue et al., (2016) overestimated the net transport through Drake Passage. Although they are referring to bottom recirculations - different from what is shown here. We will add this point.

Figure 3 caption: Why is the 1950-54 climatology used here for comparison? Because it is close to the initial conditions? Yes. Although it is also the case that the 1950-1954 climatology is likely very similar to a later climatology (see point 2 above), so it could be viewed as comparing to a "best available" long-term climatology.

L122-128: How different are the sea ice edge locations? It would be helpful to add these to the plots of Figure 5 for reference of where these anomalies are. We will add lines showing the maximum sea ice extent.

L131-133: Given that the gyres and ASC are governed by different dynamics, I strongly suggest breaking this down into an assessment of gyre strength and the ASC separately. Addressed under point 1 above.

L143: "and comparing to a similar average performed on the 1950-1954 climatology of the EN4.1.1.g10 analysis dataset". I might be missing something, but why is this time period used for comparison? Observations would be very sparse for this time period and particularly so in the Southern Ocean. Addressed under point 2 above.

L157-158: Yes, this could indicate and issue with CDW cross-shelf intrusions, but this could also be due to the westward transport of cold, fresh Weddell Sea water around the WAP mixing with CDW (making it colder & fresher). The maps of ocean velocities support this connection. This has been found in other simulations when the ASC accelerates (Beadling et al., 2022) and this mechanism has been documented as well by Morrison et al., (2023) "Weddell Sea Controls of Ocean Temperature Variability on the Western Antarctic Peninsula". You mention this below in lines 168 – 169 but it should be mentioned here or this discussion combined. We will add something about the advection of fresh water at this point.

L163-165: This would be shown nicely with a surface water mass transformation analysis (sWMT). This is also consistent with Tesdal et al., (2023) which showed that when the ASC accelerates, DSW reduces as the shelf becomes more buoyant. We agree and we plan to look at water mass transformation metrics as part of future analysis (see point 4 above).

Figures 1,3,4,5,8. It would make comparisons easier for the reader to add two additional panels to these plots of the N216-ORCA025-GM MINUS N216-ORCA025 and N216-

ORCA025-PS MINUS N216-ORCA025. It is hard as of now to discern some of the differences. We are a bit reluctant add more panels to the existing figures as there are a lot of plots already and we think that (perhaps with the exception of Fig 2) the differences between the N216-ORCA025 experiment and the two sensitivity experiments are fairly clear by eye (eg. Weddell gyre strength in Fig 1 or Amundsen Sea temperature in Fig 4). We propose to provide difference plots but to include them as supplementary figures as suggested by reviewer 2.

Figure 2 & L200-203. It looks like GM really impacts the westward along-slope flow (ASC) through the passage while the rest appears unchanged. PS appears to reduce flow everywhere (even the eastward flow in the Subantarctic Front), however reduces the eastward components more … which is why the net increases. It is hard to see visually what component of the along-slope flow is decreasing --- is this mostly coming from the bottom flow or surface intensified flow? We agree that it is hard to see the details of the impact of PS and GM in this plot – this should be clearer in the supplementary difference plot.

Figure 6: The lines for N216-eORCA025-GM and N216-eORCA025-PS are very hard to discern. Can you make one have circle markers? The dashed and the dashed-dotted are very hard to distinguish. We will experiment with alternatives to make this clearer.

L205: "The timeseries show that again, Gent-McWilliams appears to have a stronger impact than partial slip." This sentence is referring to shelf salinity, yet this is only true for the Ross. The PS West Antarctic shelve Amundsen / Bellinghausen) looks better for the PS (Figure 3) (This is ALSO true for shelf temperature as you mention below, so this would just require rewording). The timeseries for the Weddell salinity looks similar between the two. We will clarify this in the text.

All figures: Increase size of text on color bars / axes, some of these are hard to see. We will adjust these.

Figure 7:

• The top figures from Thompson et al. (2018) have a y-axis in km, but the rest are in m. This should be made to be consistent across the panels. Text on axes are also hard to read. The titles on the top and bottom also look very large compared to the other labels in the other figures in the manuscript. We will follow these suggestions.

• Figure 7: I assume that the grey shaded regions are not the models true bathymetry? The blocky-nature makes it appear that the models do not account for partial cells. I assume in reality this is more smooth? It is true that the masking in the plots does not include partial cells, and that if this were done, the bathymetry would look smoother. However, partial cells have the biggest smoothing impact in weakly sloping bathymetry. Over the shelf slope the bathymetry is still quite blocky even with partial cells.

• Figure 7: The Thompson et al., figures are conservative temperature, not potential temperature. These should be consistent between the observation panels and the model output panel. Thanks for pointing out this oversight – we will correct it.

L276: This is consistent with the feedback to meltwater Bronselaer et al., (2018) suggested i.e., slumping of isopycnals resulting in more heat delivery to shelf --- just pointing it out, but perhaps not necessary to discuss. Yes that's a good point. The mechanism described in their paper is similar to the one we are proposing here.

**Detailed comments reviewer 2**

L. 30-32: It would be appropriate to also mention the emergence of models with unstructured grid configurations, as they are an approach to overcoming the issue with affording highenough resolu:on to resolve high-la:tude eddies in global models, e.g. FESOM (Wang et al., 2013; Scholz et al., 2019), ICON (Jungclaus et al., 2022;Korn et al., 2022). We will add something about unstructured models.

L. 61-62: "A free-slip boundary condition […]" - It would be valuable to also mention what viscosity scheme is used, and potentially also the parameter settings since they likely differ between resolutions, as these also have the potential to affect the flow and thus the biases. Agreed – we will add the information about the viscosity choices.

L. 64-65.: "Diffusion of tracers along isopycnal surfaces, parameterising eddy mixing […]" – Is there any regional reduction of the parameterised eddy mixing around the equator where eddies are fully resolved, in particular in the higher resolutions? No. There is a reduction of the diffusion coefficient with reduced grid spacing at higher latitudes to avoid numerical instability.

L. 75: "with the N216 atmosphere" - Are the results consistent or at least similar if one of the other atmospheric resolutions are chosen? This would indicate that the results are more widely applicable to other coupled models, regardless of what atmospheric setup they use. We have not looked at the Southern Ocean biases in the other HighResMIP integrations in detail, but Roberts et al (2019) show that for the ACC transport, a change in atmosphere resolution makes little difference to the long-term behaviour (their Fig 18). For the SST biases there is some impact of atmosphere resolution (their Fig 7) as might be expected.

Page 3, footnote 1: Was the eddy-permitting model ever tested with no-slip? It would be useful to motivate the choice of partial-slip over no-slip in this case, and discuss how choosing a no-slip condition might have impacted the overly strong ASC. Useful references may be Penduff et al., 2007; Deremble et al., 2011; Nasser et al, 2023. The choice of free slip for the 1/4˚ model goes back to Barnier et al (2006) and Penduff et al (2007). We have not tested the 1/4˚ (or 1/12˚) model with no slip, although with

hindsight that might have been an informative thing to do. Any choice of lateral slip condition in z-level models is difficult to justify on a physical basis, and we tend to see this result as an indication that the large scale biases we are looking at may be linked to poor representation of bathymetry-flow interaction in the model.

L. 93-95: "As well as having more active gyres, the higher resolution models also have a stronger ASC [...]" - In the next paragraph, you mention the consequences of this feature in the Drake Passage, but only after you discuss the over-flattened isopycnals. This makes this part of the text feel somewhat fractured. Maybe mention the Drake Passage briefly already here, or rearrange the next paragraph to discuss the ASC behaviour before the overly flat isopycnal slopes. Following the suggestion of reviewer 1 we intend to include an extra figure showing the depth-mean currents for the top 500m. This should illustrate the circumpolar nature of the ASC in the eddying models – we will mention the westward flow in the Drake Passage when describing this figure.

L. 98-99: Klatt et al. (2005) is one of few observationally-based estimates of the Weddell Gyre strength published in recent decades. It is, however, not the only one. Reeve et al. (2019) estimate it to 32 +/- 5 Sv based on ARGO data. Older observational estimates by Farbach et al. (1991) and Yaremchuk et al. (1998) are also lower c.f. Klatt et al. Meanwhile, the transport in models ranges between at least 10-80 Sv (see e.g., Neme et al., 2021; Wang, 2013). We will mention the other observational estimates in the revised text.

L. 105-106: "The weaker ACC transport in the higher resolution models is associated with a flattening of the time-mean isopycnal slopes in the Drake Passage" - Given that particularly the ORCA025 resolution is not fully eddy-resolving in the ACC region, it is somewhat counterintuitive that there is an over-flattening of the isopycnals, which suggests too much eddy compensation. From reading the text here, it makes one wonder how that can be. As mentioned in the general comments, this counterintuitive behaviour is mentioned by the authors later in the manuscript, but should be acknowledged sooner. However, looking at the figures, it looks like the isopycnals are very steep (steeper than ORCA1) in the northern part of DP, where the core of the ACC is, and then flattened in the centre, where there is weaker (ORCA12)/counter(ORCA025) flow. In ORCA12, steeper isopycnals than in ORCA1 are also observed between -63 and -62 degN. These details should be mentioned, as they might help elucidate what is actually happening to the ACC transport. For general response see point 5 above. We will include a more detailed description of the isopycnal slopes in the Drake Passage in the revised manuscript.

L. 107: "counterflowing currents [...] associated with the Shackelton fracture zone" – The authors mention that the modelled counterflow along the southern shelf break is unrealistic c.f. Meijers et al. (2016), but they give no indication of whether counter flows at the Shackelton fracture zone correspond to observations or not. Xu et al (2020) show

complex recirculations in the Drake Passage, especially at depth, in their 1/12° model and argue that these recirculations, if realistic, are challenging to sample with observational arrays. We will make this point in the revised text.

L. 122-128: On resolution-dependent temperature biases - What are the differences in global mean surface temperature in these simulations, and compared to the observational dataset? As some of the biases in SST can stem from the atmospheric model, or results of the difference in resolution in other regions, it might be useful to make a supplementary figure where the biases are normalised to the observational global mean surface temperature. The HadGEM3 HighResMIP models tend to have cold SST biases away from the Southern Ocean and these cold biases tend to reduce as the ocean resolution is increased (Roberts et al, 2019, Fig 7) mainly due to improved representation of boundary currents and ocean heat transports. We agree that the attribution of SST biases is more complex than for some of the other biases examined in the paper because of the direct influence of the atmosphere and possible teleconnections, but given the complexity of the global picture we aren't sure that a comparison with the global mean SST bias will be very informative here.

L. 165: "in the eddy-permitting model [...] deep water formation has been suppressed" - Does this lead to (more) unrealistic open-ocean convection than in the other two model versions. In the control runs, both of the eddying models develop regular open-ocean convection after a few decades. The 1/4° model seems to be no worse than the 1/12° model in this respect.

L. 185-189: On introduction of the scale-aware G&M: What are the implications of running a resolution that to some degree allows eddy formation, but then also parametrising eddies on top of it using G&M, which can lead to "smoothing out" of the actual eddies? This should be clarified in the text. The idea is similar to that in Hallberg (2013) where we only "switch on" GM where the model fails to resolve eddies, so that we don't smooth out the eddies where the model is eddy-resolving. But of course this is difficult to do in a precise way and the whole question of how to parametrise eddies in partially-eddying models is a big research topic as we have noted in Appendix B. We will expand on this a bit in the main text.

L. 190-192: On the introduction of the partial-slip condition – As mentioned above, it would be useful to motivate why the choice was to go with partial-slip and not no-slip for this resolution. Also, based on the description of how partial slip affects the biases in ORCA025, it seems that introducing it south of 50S in ORCA12 could potentially fix the remaining biases with counter flows in the Drake Passage, and thus with cold waters being advected around the Antarctic Peninsula in this resolution as well. As discussed under point 6 above we have included the scale-aware GM and partial slip in the Southern Ocean in our standard 1/12° model and the combination of these two changes does appear to reduce the Drake Passage counterflow in this model.

L. 223-224: "the Fresh Shelf, the Dense Shelf and the Warm Shelf. In Figure 7" - It would help guide the reader's eye if it were also indicated in the figure and/or the figure caption what column exemplifies which one of these three regimes. We agree this would be clearer and will add these labels to the figure.

L. 238-239: About the V-shaped pattern of isopycnals in the higher-resolution models - I struggle to find this V-shape in the drawn isopycnals in the eddy-permittng resolution. If so, I can only see it in the two isopycnals labelled 27.5 (this might be a mistake in the labelling, as it occurs twice). We agree that the eddy-permitting model only has a shadow of the V-shape – and it also clearly fails to represent the cascade of dense water. We will update the text to make this clearer.

L. 262-265: The EN4. 1 climatology does not show the same steep ispoycnal slopes near the continent as the model in this region but, as mentioned, the observations included in the climatology are sparse. It could be useful to cite other data (not included in the EN4.1 climatology) that give indications about the isopycnal structure in the area even if those are from from a later time period. Pina-Moleno et al (2016) show observations with a strong ASC and associated front in the same sector of Antarctica. We will cite this reference here.

L. 310-320: On open-ocean polynyas - Are these events stronger/more frequent in the eddy-permittng c.f. the eddy-rich resolution, given that the latter appears to have some more capability of forming dense water on the shelves? (see also L. 165) See response to L. 165 comment.

L. 332-336: About biases along the continental slope/shelf – Here, it would be helpful to clarify which factor is the more important in reducing these biases: adding G&M or introducing the partial-slip condition. Based on the timeseries in Fig 6, they seem to be equally effective for most bias metrics, but GM seems to have a stronger impact on the Ross gyre strength and the Western Ross salinity. We will add a sentence to that effect.

Figure 5: mean SST-lines - It is unclear to me which mean SST this refers to, and as the lines are completely unlabelled, there is no indication of what temperatures the different lines represent. The lines in Fig 5 are mean SSH and are simply there to help locate the SST biases with respect to the gyres and the ACC.

Figure 9: In this figure, it might be more illustrative to show the model results as anomalies from the reanalysis data. In the other figures, observational datasets are shown in the bottom-right subpanel. It would be helpful to keep the same structure throughout the manuscript. We would prefer to keep these as plots of the plain fields rather than anomalies, in line with the approach in the other figures, but we will provide anomaly plots as a supplementary figure. We will move the JRA plot to the bottom right as suggested.

Figures overall: It could be helpful with supplementary figures showing the differences between the standard N216-ORCA025 and the other two ORCA025 versions (GS and PM) as anomalies from the standard (GS-standard, and PM-standard). We will provide the difference plots as supplementary figures. (This was also requested by reviewer 1).

References

Barnier et al (2006): https://doi.org/10.1007/s10236-006-0082-1

Good et al (2013): https://doi.org/10.1002/2013JC009067

Hallberg (2013): https://doi.org/10.1016/j.ocemod.2013.08.007

Pina-Moleno et al (2016): https://doi.org/10.1002/2015JC011594

Rignot et al (2013): https://doi.org/10.1126/science.1235798

Roberts et al (2019) : https://doi.org/10.5194/gmd-12-4999-2019

Penduff et al (2007): https://doi.org/10.5194/os-3-509-2007

Xu et al (2020): https://doi.org/10.1029/2020JC016365

---

## Author Response (AR1)

*Resolution dependence of interlinked Southern Ocean biases in global coupled HadGEM3 models.*

**Authors' response to reviewers and list of changes to revised manuscript**

We would like to thank both reviewers for taking the time to provide detailed and constructive reviews.

Here we respond to the general comments of both reviewers first and take the more detailed comments later.

**General comments**

1.  Separation of the recirculating gyre transport and ASC transport. Reviewer 1 makes the point that the dynamics of the gyre and the slope current are distinct and suggests that we provide figures for the ASC transport separately by calculating the transport over the continental slope. In the region of the gyres it is difficult to separate the ASC from the gyre, since the recirculating gyre transport impinges on the continental slope and models at these resolutions do not form very distinct jets – see for example the plots below showing cross sections of currents and depth-integrated transport densities from the MM model in the Weddell gyre.

[Figure]

In order to give some indication of the circumpolar flow distinct from the recirculating flow, we have followed the reviewer's suggestion and provided plots of the depth-mean currents in the top 500m (Fig 2 revised manuscript) to complement the plots of the streamfunction in Fig 1. We have modified the text in L105 and L277 (revised manuscript) to refer to this figure. We have also provided an extra timeseries plot in Fig 7 (revised manuscript) showing the timeseries of the counterflow at the southern boundary in the Drake Passage (as defined in Fig 2) compared to the observations of Meijers et al (2016).

2. Use of EN4 1950-1954 climatology for assessment. Both reviewers make the point that observations in the Southern Ocean for this period are extremely sparse, and therefore advise caution in the conclusions drawn by comparing the model with the EN4 climatology for this period. The EN4 analysis uses a climatology for the period 1970-2000 as a background climatology to which the solution relaxes in the absence of observations (Good et al, 2013). Given the lack of observations in the Southern Ocean, the 1950-1954 climatology is likely to be very close to this background climatology.

   We have added a footnote (to L91 of the revised manuscript) to this effect.

3. Length of integration and experimental design. Both reviewers point out that the integrations are short in the context of climate studies, and reviewer 1 questions the statement that *"the early spin up of the model can be useful in diagnosing model biases, since at this stage the model has not drifted too far from initial conditions"* because the initial state of the ocean is out of balance with the greenhouse gas forcing being used and there will be an adjustment associated with this. These experiments follow the HighResMIP protocol (Haarsma et al., 2016), in particular the *spinup-1950* and *control-1950* experiments, in which the model is initialised with "1950s" initial conditions and spun up with 1950s greenhouse gas forcing. As stated by Roberts et al (2019), the nominal spin up for these integrations is very short (30 years) due to computational constraints. As noted in point 2 above the "1950s" initial conditions are likely more similar to a later state of the ocean, so there is an imbalance between initial conditions and the forcing, but the imbalance will not be as great as for a pre-industrial spin up integration. Still it is clear that the model is adjusting in the first decades of the spin up and nowhere near an equilibrated state. However from the timeseries in Fig 6 it seems that there is a fast initial adjustment over the first 2-3 decades after which the three models reach very different states, and these differences in many cases persist for multi-centennial timescales. (We have only shown the first 100 years in Fig 6 because of the length of the sensitivity experiments). In particular, the biases in the MM model appear to be very persistent, so we believe we can learn something about them by studying the initial adjustment period.

   We have rewritten the opening paragraph of Section 2.2 (L87-95) to emphasise that we are following the HighResMIP protocol and argue that we can learn something about the impact of resolution on the model biases from these relatively short integrations in the spirit of Roberts et al 2019.

4. Dense water formation and export. Reviewer 1 suggests some analysis of dense water formation and export from the shelf and AABW export to shed light on the water mass biases on the shelf and possible spurious mixing in the interior. We agree that the models' representation of dense water formation and export could well be relevant to the formation of the biases studied in the paper. From the cross sections in Fig 7 it seems clear that the export of dense water from the shelf in the Weddell Sea is only captured by the 1/12˚ model. We plan to do a systematic study of the dense water formation in the different models looking at the water mass transformation on the shelf, the export over the shelf break and the time evolution of the reservoir of AABW, but we would prefer to present this in another paper.

We have discussed the potential importance of the models' representation of dense water formation and export in the final paragraph of the Conclusions (L401-410 revised manuscript) and stated that it will be the subject of future work.

5. Explanation for isopycnal slumping in the open ocean. Reviewer 2 says that the slumping of isopycnals across the ACC in the medium resolution model is counter intuitive and that this is not addressed very prominently in the paper, with a tentative explanation only appearing in the last paragraph of the Conclusions.

We agree and we have followed the reviewer's suggestion of emphasising the counter-intuitive nature of the result in Section 4.2 of the revised manuscript. We have renamed this section to emphasise the discussion on slumping isopycnals. The discussion of the possible link to spurious mixing and the loss of AABW has been moved from the Conclusions to this section.

6. Inclusion of partial slip and scale-aware Gent-McWilliams in 1/12˚ model. Reviewer 2 asks if we tested partial slip or scale-aware Gent-McWilliams in the 1/12˚ model. We have included the combination of partial slip and scale-aware GM in our standard 1/12˚ model configuration, but due to computational expense we only have a clean test of the impact in a forced ocean-ice configuration. In this test, the combined effect of partial slip and scale-aware GM is still positive in the sense that it increases the ACC strength and reduces the gyres, but the impact is not as great as it is for the 1/4˚ model. This might be expected because the increased resolution at 1/12˚ means that the GM coefficient will be non-zero over a smaller area than for the 1/4˚.

We have mentioned these results with the 1/12˚ model in the Conclusions (L390-393 of the revised manuscript).

**Detailed comments reviewer 1**

L16: "Antarctic Overturning Circulation". I think stating "between the near surface and deep ocean via the formation of Dense Shelf Water (DSW), Antarctic Bottom Water (AABW), and mid-depth ocean via mode and intermediate water formation" would work better here. If you are linking to the Southern Ocean being critical to the climate system - also mentioning the mode & int water formation is important as this is where all the heat and carbon is going.

We have made the suggested change.

L23: Capitalize C D and W in "Circumpolar Deep Water".

Done.

L24: Also, there are a lot of warm biases in 1-degree CMIP-class models (Beadling et al., 2020), not sure there is a definitive link to high resolution models being warmer? For example, in some high-res simulations we actually see warm biases improve with finer nominal grid spacing.

Agreed – the statement about the resolution dependence may have been a Met Office centric point of view. We have changed this to simply state that warm biases in the Southern Ocean are a common problem in CMIP models (L25 revised manuscript).

L25: It would be useful to cite Hallberg 2013 here which shows this limitation very well. Hallberg, R. (2013). Using a resolution function to regulate parameterizations of oceanic mesoscale eddy effects. Ocean Modelling, 72, 92–103. https://doi.org/10.1016/j.ocemod.2013.08.007.

We have added a reference to Hallberg (2013) together with the reference to Hewitt (2022) in L28 revised manuscript.

L50-53: This paragraph is probably not necessary.

We think it is useful to have a very brief overview of the structure of the paper and have left it in the revised manuscript, but are happy to remove this if the editor prefers.

L56-57: These two sentences appear as separate from the paragraph below. I suggest combining these into the paragraph below.

We have combined these two paragraphs.

L60: "and partial cells (Barnier et al., 2006; Adcroft et al., 1997) allowed next to topography." -> "and partial cells at the ocean bottom to better represent bathymetry (Barnier et al., 2006; Adcroft et al., 1997)."

That reads better. We have made that change.

L67: "Cavities under ice shelves are closed and the output of basal melt water at the ice shelf front parametrised as described in Mathiot et al. (2017)." This is interesting, so this model represents "ice shelf melt"? Is this just based on some threshold of solid precip over the Antarctic continent? Could you elaborate on this? I assume this does not imply there are realistic melt rates?

The distribution of melt water input around the continent is based on Rignot et al (2013), but for these experiments the overall magnitude of melt water plus iceberg calving is scaled to equal the total precipitation over the Antarctica at each timestep, ie. An assumption that the total mass of ice over the continent is constant. We have expanded the description in the text to explain this (L71-74 revised manuscript).

L83-84: I suggest to briefly elaborate on the term "model drift" here for readers that are not familiar.

We have expanded this sentence to make it more explanatory – L87-89 revised manuscript.

L84-85: "However the early spin up of the model can be useful in diagnosing model biases, since at this stage the model has not drifted too far from initial conditions." ß I am not sure I agree with this statement, assuming the ocean is starting from a present-day climatology (say WOA13 or WOA18) and a pre-industrial atmosphere, this early stage is an unrealistic climate state and an assessment of realism (i.e., biases relative to observed) is better made once the model has been able to achieve its own equilibrium (or better yet,reached that equilibrium and forced with observed climate forcings; i.e., the historical simulation). I tend to think of the spin-up stage as the adjustment stage that we don't want to consider when doing assessments against observations. I suggest rewording this or expanding on your reasoning here.

See response under General Commets point 3 above. We have rewritten this paragraph to make our approach clearer.

L93-101: As you note, the gyres and ASC transports merge into one another particularly in the Weddell, so it is hard to discern these from one another in the current figures. I would suggest an additional plot of the upper 1000 m speed (or upper 500 m speed) to see the differences in the strength and location of the ASC. L93-101: As you note, the gyres and ASC transports merge into one another particularly in the Weddell, so it is hard to discern these from one another in the current figures. I would suggest an

additional plot of the upper 1000 m speed (or upper 500 m speed) to see the differences in the strength and location of the ASC.

See response to General Comments point 1. We have provided maps of the depth-integrated currents over the top 500m in Figure 2 of the revised manuscript to complement the plots of the streamfunction in Fig 1 and we have updated the text in L105 and L277 (revised manuscript) to refer to the new figure.

L102: "net eastward transport" this wording is confusing, there is eastward and westward flow through Drake Passage, should this just say "net" to avoid confusion?

We have used "net transport" instead of "net eastward transport" throughout the revised manuscript.

L103-104: It is worth mentioning that this is exceptionally weak even compared to earlier / other estimates (Cunningham et al., 2003; Griesel et al., 2012; Meijers et al., 2012; Koenig et al., 2014; Firing et al., 2011; Xu et al., 2020).

We have added this point (L114-116 revised manuscript).

L108: "These counterflows significantly reduce the net eastward transport." I would remove the word eastward after net and just say "these westward counterflows significantly reduce the total net transport".

We have used "net transport" instead of "net eastward transport" to be consistent with the rest of the manuscript.

L109-111: It is worth mentioning that Xu et al., (2020) also shows net westward flow at depth - this is why the authors argue that Donohue et al., (2016) overestimated the net transport through Drake Passage. Although they are referring to bottom recirculations - different from what is shown here.

In fact, the 1/12 degree model discussed here has bottom-intensified recirculations similar to those described in Xu et al. but the ¼ degree model has very barotropic recirculations. We have added this point to the paragraph (L123-125 revised manuscript).

Figure 3 caption: Why is the 1950-54 climatology used here for comparison? Because it is close to the initial conditions?

Yes. Although it is also the case that the 1950-1954 climatology is likely very similar to a later climatology (see point 2 above), so it could be viewed as comparing to a "best available" long-term climatology.

L120: "partially eroded to the east of this region in the eddy-rich model" - swap "in the east" to "to the east". Also, this is likely due to the delivery of cold, fresh Weddell Sea water to the WAP.

We have made this change.

L122-128: How different are the sea ice edge locations? It would be helpful to add these to the plots of Figure 5 for reference of where these anomalies are.

We have added lines showing the September sea ice extent to the plots in Figure 5 and updated the corresponding text in L141, and L241-242.

L131-133: Given that the gyres and ASC are governed by different dynamics, I strongly suggest breaking this down into an assessment of gyre strength and the ASC separately.

As noted under point 1 above it is difficult to separate the gyre transport and ASC transport in the region of gyres. We have added a timeseries of the magnitude of the westward counterflow in the Drake Passage to Figure 6, to give an indication of the slope transport not associated with recirculating gyres.

L133: remove "eastward" before "net" due to comments earlier.

Done.

L131-135: These series of sentences sound a bit choppy as they are currently written with all starting with "We", "We", "We".

We have rewritten these sentences.

L134: Remove "deep" before "fields below 400 m" as "fields below 400 m" imply they are deep.

Done.

L143: "and comparing to a similar average performed on the 1950-1954 climatology of the EN4.1.1.g10 analysis dataset". I might be missing something, but why is this time period used for comparison? Observations would be very sparse for this time period and particularly so in the Southern Ocean.

Addressed under General Comments, point 2 above.

L157-158: Yes, this could indicate and issue with CDW cross-shelf intrusions, but this could also be due to the westward transport of cold, fresh Weddell Sea water around the WAP mixing with CDW (making it colder & fresher). The maps of ocean velocities support this connection. This has been found in other simulations when the ASC accelerates (Beadling et al., 2022) and this mechanism has been documented as well by Morrison et al., (2023) "Weddell Sea Controls of Ocean Temperature Variability on the Western Antarctic Peninsula". You mention this below in lines 168 – 169 but it should be mentioned here or this discussion combined.

We have mentioned the possible advection of fresh water around the Antarctic Peninsula at this point (L177-178 revised manuscript).

L163-165: This would be shown nicely with a surface water mass transformation analysis (sWMT). This is also consistent with Tesdal et al., (2023) which showed that when the ASC accelerates, DSW reduces as the shelf becomes more buoyant.

We agree and we plan to look at water mass transformation metrics as part of future analysis (see General Comments, point 4 above). We have added this point in the last paragraph of the Conclusions in the revised manuscript.

L180: Add "ocean" before circulation.

Done.

L183: Should you add "and associated treatment of mesoscale eddies" after "ocean resolution", to be clear this is not just due to horizontal grid spacing alone?

Yes that is a good point. Done.

L201: remove "eastward" when referring to the net transport.

Done.

Figures 1,3,4,5,8. It would make comparisons easier for the reader to add two additional panels to these plots of the N216-ORCA025-GM MINUS N216-ORCA025 and N216-ORCA025-PS MINUS N216-ORCA025. It is hard as of now to discern some of the differences.

In order to avoid cluttering the original figures we have followed the second reviewer's suggestion to include difference plots showing the impact of Gent-McWilliams and partial slip on the temperature and salinity fields in supplementary figures as an appendix – Figures A1-A4 in the revised manuscript. We have updated the text to refer to these in L217-218, 230, 233 and 240 and 302 of the revised manuscript.

Figure 2 & L200-203. It looks like GM really impacts the westward along-slope flow (ASC) through the passage while the rest appears unchanged. PS appears to reduce flow everywhere (even the eastward flow in the Subantarctic Front), however reduces the eastward components more … which is why the net increases. It is hard to see visually what component of the along-slope flow is decreasing --- is this mostly coming from the bottom flow or surface intensified flow?

The details of the impact of PS and GM in the Drake Passage are complicated. We have rewritten the end of this paragraph (L225-228 revised manuscript) to try to better capture the differences.

Figure 6: The lines for N216-eORCA025-GM and N216-eORCA025-PS are very hard to discern. Can you make one have circle markers? The dashed and the dashed-dotted are very hard to distinguish.

We have used dotted lines instead of dash-dotted lines for N216-ORCA025-PS. Hopefully this is clearer.

L204: "The fresh biases on the shelves are reduced, with some recovery of the HSSW in the western Weddell Sea and western Ross Sea (Figure 3)" - The shelves in the South Indian sector appear relatively unchanged too.

The difference plots (Figure A1) show some reduction of the fresh bias in the Indian sector as well.

L205: "The timeseries show that again, Gent-McWilliams appears to have a stronger impact than partial slip." This sentence is referring to shelf salinity, yet this is only true for the Ross. The PS West Antarctic shelve Amundsen / Bellinghausen) looks better for the PS (Figure 3) (This is ALSO true for shelf temperature as you mention below, so this would just require rewording). The timeseries for the Weddell salinity looks similar between the two.

We have expanded the text in L230-238 to describe the results in more detail and include the reviewer's point that PS has a stronger impact than GM in the Amundsen Sea for both temperature and salinity.

All figures: Increase size of text on color bars / axes, some of these are hard to see.

We have increased the font sizes for axis and color bar labels on all figures.

Figure 7:

• The top figures from Thompson et al. (2018) have a y-axis in km, but the rest are in m. This should be made to be consistent across the panels. Text on axes are also hard to read. The titles on the top and bottom also look very large compared to the other labels in the other figures in the manuscript.

We have increased the font sizes for axes and colour bars, and reduced the font sizes for the column headings. We have labelled the depth axis in km for all plots.

• Figure 7: I assume that the grey shaded regions are not the models true bathymetry? The blocky-nature makes it appear that the models do not account for partial cells. I assume in reality this is more smooth?

It is true that the masking in the plots does not include partial cells, and that if this were done, the bathymetry would look smoother. However, partial cells have the biggest smoothing impact in weakly sloping bathymetry. Over the shelf slope the bathymetry is still quite blocky even with partial cells.

• Figure 7: The Thompson et al., figures are conservative temperature, not potential temperature. These should be consistent between the observation panels and the model output panel.

This was an oversight – we have now plotted the model results as Conservative Temperature to match the Thompson et al figures.

L220: "The properties of the shelf water are controlled partly by local surface fluxes and partly by the exchange of water with the open ocean across the shelf break" -> "The properties of the shelf water are controlled partly by local surface fluxes and associated water mass transformations, and partly by the exchange of water with the open ocean across the shelf break"

We have made this change.

L223-226: This paragraph seems short for a stand-alone paragraph, consider merging with the one below.

We would prefer to keep this paragraph separate (even though it is short) because then we have a separate paragraph for each of the Fresh Shelf, Dense Shelf and Warm Shelf cases.

L229: This statement needs a citation: "these incursions are likely to only happen occasionally as tidal or eddy driven fluctuations of the front position onto the shelf."

We have added citations for Wang et al 2013 (action of tides) and Goddard et al 2017 (action of eddies). L261 revised manuscript.

L237: Cite Thompson et al., (2018) after this statement: "the observed structure is more complex, with a V-shaped pattern of isopycnals associated with the incursion of CDW onto the shelf and its transformation and export as Dense Shelf Water (DSW)".

It's not clear that this is necessary, given that this whole section is comparing to the observations shown in the Thompson et al paper?

L255 (and throughout): Degree sign missing between number and letter for longitude (this is also true for latitudes in the text).

We have corrected this throughout the manuscript.

L247: Add comma after "in general".

Done.

L268-270: Paragraph is short – suggest combining.

Done.

L276: This is consistent with the feedback to meltwater Bronselaer et al., (2018) suggested i.e., slumping of isopycnals resulting in more heat delivery to shelf --- just pointing it out, but perhaps not necessary to discuss.

Yes that's a good point. The mechanism described in their paper is similar to the one we are proposing here.

L281: Add "ocean" before flow.

Done.

L284: Should this be "wind stress curl", not "wind curl"?

Yes we have changed this.

L285: What does "medium resolution" imply in km?

N216 is roughly equivalent to 60km resolution. We have added a footnote to this effect.

**Detailed comments reviewer 2**

L. 1-2: "eddy-permittng ocean resolution" -> "eddy-permitting ocean resolution without additional eddy parameterisation"

This isn't strictly correct because the eddy-permitting (and eddy-rich) models both include isopycnal diffusion which is supposed to represent the action of eddies. The inclusion of isopycnal diffusion, but no Gent-McWilliams type parametrisation in eddying models is common practice (if slightly strange) and we explicitly state that we are testing the use of a kind of Gent-McWilliams parametrisation later in the abstract, so we would prefer to leave this as just "eddy-permitting ocean resolution"

L. 8: "unresolved eddy processes or the representation of bathymetric drag" -> and/or, since both can be (and are) causing issues at the same time.

We have included this change.

L. 9: Already here in the abstract, the authors mention the shallower isopycnal slopes of the eddy-permittng resolution, and it immediately caught my attention as being counterintuitive. Hence, I would have liked to see this acknowledged earlier in the paper itself.

See under General Comments point 5 above. We have discussed the counter intuitive nature of this result and the possible explanation in Section 4.2 in the revised manuscript. We have modified the title of this section to make the discussion of the slumping isopycnals more prominent.

L. 30-32: It would be appropriate to also mention the emergence of models with unstructured grid configurations, as they are an approach to overcoming the issue with affording high enough resolution to resolve high-latitude eddies in global models, e.g. FESOM (Wang et al., 2013; Scholz et al., 2019), ICON (Jungclaus et al., 2022;Korn et al., 2022).

We have added a footnote about unstructured grid models to L33 revised manuscript.

L. 61-62: "A free-slip boundary condition […]" - It would be valuable to also mention what viscosity scheme is used, and potentially also the parameter settings since they likely differ between resolutions, as these also have the potential to affect the flow and thus the biases.

We have added a new table (Table 1 in the revised manuscript) with the viscosity and isopycnal diffusion parameters settings and referred to this in the main text.

L. 64-65.: "Diffusion of tracers along isopycnal surfaces, parameterising eddy mixing […]" – Is there any regional reduction of the parameterised eddy mixing around the equator where eddies are fully resolved, in particular in the higher resolutions?

No. There is a reduction of the diffusion coefficient with reduced grid spacing at higher latitudes to avoid numerical instability. We have added a note to this effect in the new Table 1.

L. 75: "with the N216 atmosphere" - Are the results consistent or at least similar if one of the other atmospheric resolutions are chosen? This would indicate that the results are more widely applicable to other coupled models, regardless of what atmospheric setup they use.

We have not looked at the Southern Ocean biases in the other HighResMIP integrations in detail, but Roberts et al (2019) show that for the ACC transport, a change in atmosphere resolution makes little difference to the long-term behaviour (their Fig 18). For the SST biases there is some impact of atmosphere resolution (their Fig 7) as might be expected. We have added two sentences about the cross-resolution results of Roberts et al in the Conclusions (L386-390 of the revised manuscript).

Page 3, footnote 1: Was the eddy-permitting model ever tested with no-slip? It would be useful to motivate the choice of partial-slip over no-slip in this case, and discuss how choosing a no-slip condition might have impacted the overly strong ASC. Useful references may be Penduff et al., 2007; Deremble et al., 2011; Nasser et al, 2023.

The choice of free slip for the 1/4˚ model goes back to Barnier et al (2006) and Penduff et al (2007). We have not tested the 1/4˚ (or 1/12˚) model with no slip, although with hindsight that might have been an informative thing to do. Any choice of lateral slip condition in z-level models is difficult to justify on a physical basis, and we tend to see this result as an indication that the large scale biases we are looking at may be linked to poor representation of bathymetry-flow interaction in the model.

L. 93-95: "As well as having more active gyres, the higher resolution models also have a stronger ASC […]" - In the next paragraph, you mention the consequences of this feature in the Drake Passage, but only after you discuss the over-flattened isopycnals. This makes this part of the text feel somewhat fractured. Maybe mention the Drake Passage

briefly already here, or rearrange the next paragraph to discuss the ASC behaviour before the overly flat isopycnal slopes.

Following the suggestion of reviewer 1 we have included an extra figure (Fig 2 in the revised manuscript) showing the depth-mean currents for the top 500m. This illustrates the circumpolar nature of the ASC in the eddying models. We have mentioned the westward flow in the Drake Passage when discussing the strong ASC.

L. 98-99: Klatt et al. (2005) is one of few observationally-based estimates of the Weddell Gyre strength published in recent decades. It is, however, not the only one. Reeve et al. (2019) estimate it to 32 +/- 5 Sv based on ARGO data. Older observational estimates by Farbach et al. (1991) and Yaremchuk et al. (1998) are also lower c.f. Klatt et al. Meanwhile, the transport in models ranges between at least 10-80 Sv (see e.g., Neme et al., 2021; Wang, 2013).

We have added a footnote to L109 (revised manuscript) referring to the Yaremchuk et al and Reeve et al estimates and justifying our choice of the Klatt et al. as being the most comparable observational estimate to our Weddell gyre strength metric.

L. 105-106: "The weaker ACC transport in the higher resolution models is associated with a flattening of the time-mean isopycnal slopes in the Drake Passage" - Given that particularly the ORCA025 resolution is not fully eddy-resolving in the ACC region, it is somewhat counterintuitive that there is an over-flattening of the isopycnals, which suggests too much eddy compensation. From reading the text here, it makes one wonder how that can be. As mentioned in the general comments, this counterintuitive behaviour is mentioned by the authors later in the manuscript, but should be acknowledged sooner. However, looking at the figures, it looks like the isopycnals are very steep (steeper than ORCA1) in the northern part of DP, where the core of the ACC is, and then flattened in the centre, where there is weaker (ORCA12)/counter(ORCA025) flow. In ORCA12, steeper isopycnals than in ORCA1 are also observed between -63 and -62 degN. These details should be mentioned, as they might help elucidate what is actually happening to the ACC transport.

For response to the point about slumping of isopycnals see General Comments, point 5 above. We have included a more detailed description of the isopycnal slopes in the Drake Passage in the revised manuscript (L117-123).

L. 107: "counterflowing currents […] associated with the Shackelton fracture zone" – The authors mention that the modelled counterflow along the southern shelf break is unrealistic c.f. Meijers et al. (2016), but they give no indication of whether counter flows at the Shackelton fracture zone correspond to observations or not.

Xu et al (2020) show complex recirculations in the Drake Passage, especially at depth, in their 1/12˚ model and argue that these recirculations, if realistic, are challenging to

sample with observational arrays. We have made this point in the revised text (L123-125).

L. 122-128: On resolution-dependent temperature biases - What are the differences in global mean surface temperature in these simulations, and compared to the observational dataset? As some of the biases in SST can stem from the atmospheric model, or results of the difference in resolution in other regions, it might be useful to make a supplementary figure where the biases are normalised to the observational global mean surface temperature.

The HadGEM3 HighResMIP models tend to have cold SST biases away from the Southern Ocean and these cold biases tend to reduce as the ocean resolution is increased (Roberts et al, 2019, Fig 7) mainly due to improved representation of boundary currents and ocean heat transports. We agree that the attribution of SST biases is more complex than for some of the other biases examined in the paper because of the direct influence of the atmosphere and possible teleconnections, but given the complexity of the global picture we aren't sure that a comparison with the global mean SST bias will be very informative here.

L. 132: "observational estimates of Klatt et al. (2005)" – as this is not the only observational estimate of the Weddell Gyre strength (see L. 98-99), this may need to be modified, or at least motivated why this particular estimate is used for comparison.

See response to L98-99 above.

L. 165: "in the eddy-permitting model […] deep water formation has been suppressed" - Does this lead to (more) unrealistic open-ocean convection than in the other two model versions.

In the control runs, both of the eddying models develop regular open-ocean convection after a few decades. The 1/4˚ model seems to be no worse than the 1/12˚ model in this respect.

L. 168-169: Cold, fresh water advecting around the Antarctic peninsula in the eddy-rich model suggests the ASC and is too strong through the Drake Passage also in this model version.

We have added a note about the possible link between the westward flow in the Drake Passage and the cold, fresh biases in the Amundsen Sea. (L177-178 of revised manuscript).

L. 185-189: On introduction of the scale-aware G&M: What are the implications of running a resolution that to some degree allows eddy formation, but then also parametrising eddies on top of it using G&M, which can lead to "smoothing out" of the actual eddies? This should be clarified in the text.

The idea is similar to that in Hallberg (2013) where we only "switch on" GM where the model fails to resolve eddies, so that we don't smooth out the eddies where the model is eddy-resolving. But of course this is difficult to do in a precise way and the whole question of how to parametrise eddies in partially-eddying models is a big research topic as we have noted in Appendix B. We have modified the main text slightly to emphasise that we are only turning on GM where eddies are unresolved (in some sense) and to emphasise that we are testing quite a simple idea compared to some schemes in the literature (L206-212 revised manuscript).

L. 190-192: On the introduction of the partial-slip condition – As mentioned above, it would be useful to motivate why the choice was to go with partial-slip and not no-slip for this resolution. Also, based on the description of how partial slip affects the biases in ORCA025, it seems that introducing it south of 50S in ORCA12 could potentially fix the remaining biases with counter flows in the Drake Passage, and thus with cold waters being advected around the Antarctic Peninsula in this resolution as well.

As discussed under point 6 above we have tested scale-aware GM and partial slip in the Southern Ocean in a forced integration of the 1/12˚ model and the combination of these two changes does appear to reduce the Drake Passage counterflow in this model. We have added a note along these lines in the Conclusions (L391-394 of revised manuscript).

L. 223-224: "the Fresh Shelf, the Dense Shelf and the Warm Shelf. In Figure 7" - It would help guide the reader's eye if it were also indicated in the figure and/or the figure caption what column exemplifies which one of these three regimes.

We have added these labels to the figure.

L. 238-239: About the V-shaped pattern of isopycnals in the higher-resolution models - I struggle to find this V-shape in the drawn isopycnals in the eddy-permittng resolution. If so, I can only see it in the two isopycnals labelled 27.5 (this might be a mistake in the labelling, as it occurs twice).

We agree that the eddy-permitting model only has a shadow of the V-shape – and it also clearly fails to represent the cascade of dense water. We have updated the text to qualify our statement that the V-shape is present in both eddying models (L270).

L. 262-265: The EN4. 1 climatology does not show the same steep ispoycnal slopes near the continent as the model in this region but, as mentioned, the observations included in the climatology are sparse. It could be useful to cite other data (not included in the EN4.1 climatology) that give indications about the isopycnal structure in the area even if those are from from a later time period.

Pena-Moleno et al (2016) show observations with a strong ASC and associated front in the same sector of Antarctica. We have cited this reference in the revised text (L297 revised manuscript).

L. 310-320: On open-ocean polynyas - Are these events stronger/more frequent in the eddy-permittng c.f. the eddy-rich resolution, given that the latter appears to have some more capability of forming dense water on the shelves? (see also L. 165)

See response to L165 comment.

L. 332-336: About biases along the continental slope/shelf – Here, it would be helpful to clarify which factor is the more important in reducing these biases: adding G&M or introducing the partial-slip condition.

We have added a sentence about the relative impact of GM and PS in the first paragraph of the Conclusions (L371-373 revised manuscript).

Figure 1 (caption): See comments to L. 98-99 and L. 132 regarding Klatt et al. (2005)

See response to L98-99.

Figure 5: mean SST-lines - It is unclear to me which mean SST this refers to, and as the lines are completely unlabelled, there is no indication of what temperatures the different lines represent.

The lines in Fig 5 are mean SSH and are simply there to help locate the SST biases with respect to the gyres and the ACC. We have expanded the caption to make this point. At the request of reviewer 1 we have also added a dashed line showing the maximum ice extent to these plots.

Figure 9: In this figure, it might be more illustrative to show the model results as anomalies from the reanalysis data. In the other figures, observational datasets are shown in the bottom-right subpanel. It would be helpful to keep the same structure throughout the manuscript.

The main purpose of including this figure is to argue that the large scale magnitudes and patterns of the wind forcing are very similar between the different model runs and so the differences we see in the model response must arise principally from differences in the ocean model. We think that the plots of the fields presented here illustrate that basic point and we're not sure that showing anomalies against JRA would add much. We have moved the JRA plot to the bottom right in line with the other figures as suggested.

Figures overall: It could be helpful with supplementary figures showing the differences between the standard N216-ORCA025 and the other two ORCA025 versions (GS and PM) as anomalies from the standard (GS-standard, and PM-standard).

We have provided the difference plots for the temperatures and salinities between the GM and PS test and the control as appendix figures and referenced these from the main text. (This was also requested by reviewer 1).

References

Barnier et al (2006): https://doi.org/10.1007/s10236-006-0082-1

Good et al (2013): https://doi.org/10.1002/2013JC009067

Hallberg (2013): https://doi.org/10.1016/j.ocemod.2013.08.007

Pina-Moleno et al (2016): https://doi.org/10.1002/2015JC011594

Rignot et al (2013): https://doi.org/10.1126/science.1235798

Roberts et al (2019) : https://doi.org/10.5194/gmd-12-4999-2019

Penduff et al (2007): https://doi.org/10.5194/os-3-509-2007

Xu et al (2020): https://doi.org/10.1029/2020JC016365

---

## Referee Report (RR1)

**General comments**

After the first round of comments on the manuscript, which the authors addressed in a satisfactory manner, the manuscript has been improved accordingly, and I only see very minor remaining issues. The only one of my new comments that require a response from the authors is that for Fig. 6. As soon as the authors have justified their choice of reference for the bias analysis in that figure, or changed to using the same reference as in the bottom temperature analysis, I recommend the manuscript for publication. I congratulate the authors on a nice piece of work.

**Specific comments**

Table 1. I appreciate that the authors have provided the details of the parameter choices for lateral viscosity and isopycnal tracer diffusion, as this allows for some further comparison with other modelling studies and between the simulations themselves. The coefficient value for the bilaplacian viscosity, $v$, can be indicated by a targeted characteristic current velocity, $U_c$, at the grid size of the selected resolution, $\Delta x$, where

$$v = \frac{1}{12} U_c \, \Delta x^3 \qquad (1)$$

Hence, we can also infer the characteristic velocity from the selected coefficient value $v$. A typical value for $U_c$ would be about 10 cm/s. (Gurvan Madec, pers. comm.)  At the equator, 1/4º and 1/12º resolution correspond to roughly 28 and 9.3 km, respectively. The selected parameter values for $v$ thereby yield characteristic velocities of 8 and 19 cm/s. While 8 cm/s falls in the expected realm, the 19 cm/s for 1/12º seems high. It is possible that reducing the coefficient further for the 1/12º version of the model could help to further improve its performance, as this would likely act to slow down the ASC. I base this assumption on similar experiments with 1/10º NEMO-AGRIF where a reduction of the bilaplacian viscosity coefficient to reduce $U_c$ resulted in a less vigorous ASC (Ödalen et al., in prep.). I do not expect the authors to change the viscosity coefficient setting in the experiments for the present manuscript, but it could be worth noting that the setting might still be a bit too high and that this could be one source of bias for the 1/12º simulation that does not exist in the 1/4º simulation.

Fig. 6. Please justify why ESA-CCI-SST is used for the bias analysis, instead of the EN4.1 reanalysis that is used as a reference for initialisation, and for bottom temperature comparison.

**Minor comments**
L. 140. Typo: Kerguelan → Kerguelen

L. 141. To avoid starting two sentences in a row with "The warm bias", which reads a bit repetitive, the first sentence could be rearranged, so that it starts with "In this model, the warm bias…", thus removing "in this model" from the end of the sentence.

L. 239: "are reduced in the Indian Ocean sector" (in is missing in the manuscript text)

---

## Author Response (AR2)

*Resolution dependence of interlinked Southern Ocean biases in global coupled HadGEM3 models.*

**Authors' response to reviewers (second round)**

Reviewer 2 makes two further points:

1. Using a scaling analysis it appears that the biharmonic viscosity coefficient is somewhat high in the 1/12 degree model compared to the ¼ deg model, and she suggests that a reduction in the coefficient might (counter-intuitively) reduce the strength of the ASC. This is an interesting point and we look forward to reading about her experience in the upcoming paper. We haven't commented on this in the revised manuscript (partly because her paper is not available to reference yet).

2. She requests an explanation of why we chose to validate SST against satellite data rather than against the EN4.1 reanalysis used for the subsurface temperature fields. We have added a sentence to the caption of Fig 6 and also made the point that near-surface temperature anomalies against the satellite data and against EN4.1 are quite similar.

3. We have made the minor corrections/improvements that she suggests.

We have checked the figures against the online colour blindness checker and think that the figures (particularly the timeseries in Fig 7) are sufficiently decipherable under the different conditions.